# WEIGHT CLIPPING FOR ROBUST CONFORMAL INFERENCE UNDER UNBOUNDED COVARIATE SHIFTS

## ABSTRACT

Conformal prediction (CP) provides powerful, distribution-free prediction sets, but its guarantees rely on the exchangeability of training and test data, which is often violated in practice due to covariate shifts. While weighted conformal prediction (WCP) is designed to handle such shifts, it can suffer from significant undercoverage when the density ratio between the distributions is unbounded and/or must be learned. This is because of both overfitting in learning the density ratio, and high variance in estimating the nonconformity score threshold. To address this, we introduce clipped least-squares importance fitting (CLISF) as a reduced-variance method for density ratio estimation. Specifically, we show that density ratios learned using CLISF, when plugged into WCP, have bounded expected undercoverage. Furthermore, we show that the undercoverage can be corrected by running WCP with a slightly inflated coverage target; crucially, we are able to estimate the required level of inflation from the data. We provide the first theoretical guarantees for weight clipping in conformal inference, achieving dataset-conditional coverage with a sample complexity that does not blow up with the higher moments of the true density ratio—a key limitation of prior work. We verify our results on real-world benchmarks and synthetic data.

## 1 INTRODUCTION

Predictive algorithms are essential tools in medicine, finance, and the sciences, used to forecast outcomes and quantify uncertainty. Conformal prediction (CP) (Vovk et al., 2005) uses a calibration set $\mathcal{D} = \{(X_i, Y_i)\}_{i=1}^m$ to construct prediction sets $C(x)$ that contain the true outcome $y$ with a user-specified probability, $1 - \alpha$. A standard guarantee is *expected marginal coverage*, where $\Pr_{\mathcal{D}, X, Y}[Y \in C(X)] \geq 1 - \alpha$, averaging over both the calibration and test data. A stronger guarantee is *dataset-conditional marginal coverage*, which requires that for a *given* calibration set, the coverage probability $\Pr_{X,Y}[Y \in C(X)]$ is at least $1 - \alpha$, holding with high probability $(1 - \delta)$ over the draw of $\mathcal{D}$. Split conformal prediction (Papadopoulos et al., 2002) is a straightforward method to achieve these guarantees which requires exchangeability of calibration and test data.

However, the exchangeability assumption is often violated in practice due to covariate shifts, where the marginal covariate distributions change between training and test sets ($P_X \neq Q_X$), while the conditional label distribution remains invariant ($P(Y|X) = Q(Y|X)$). A standard approach to handle this is weighted conformal prediction (WCP) (Tibshirani et al., 2019), which reweights the calibration samples according to an estimate of the density ratio $w^*(x) = dQ_X/dP_X$. However, WCP can fail dramatically when this density ratio is unbounded or must be learned. First, unbounded ratios lead to high-variance estimates of the coverage threshold and greatly reduce the "effective sample size" (Tibshirani et al., 2019). Second, for an estimated ratio $\hat{w}$, Lei & Candès (2021) bound the (expected) undercoverage by $\mathbb{E}_P[|\hat{w}(X) - w^*(X)|]$. However, to guarantee this quantity is small is challenging, as generalization bounds generally fail when the error functions have bad higher moments. Consider the following motivating example:

**Example 1.** Fix a dimension $d \in \mathbb{N}$, radius $r \in (0, 1)$, and mixture weight $\theta \in (0, 1)$. Define the input space $\mathcal{X} = [0, 1]^d$ and label space $\mathcal{Y} = [0, 1]$. Define $\mathcal{B}$ to be the ball $\{x \in \mathcal{X} : \|x\|_\infty \leq r\}$. Define the train distribution $P$ to be uniform over $\mathcal{X} \times \mathcal{Y}$. Define the test distribution $Q = (1 -$

$\theta)P + \theta S$, where $S$ is uniform over $\mathcal{B} \times \mathcal{Y}$. Define the nonconformity score to be $s(x, y) = \|x\|_\infty$.

It can be checked that $\mathrm{TV}(P, Q) = \theta(1 - r^d)$ and $w^*(x) = \begin{cases} 1 - \theta + \theta/r^d, & x \in \mathcal{B} \\ 1 - \theta, & x \notin \mathcal{B} \end{cases}$.

Here, the total variation between $P$ and $Q$ is small, yet the density ratio and its higher moments are unbounded as $r \to 0$. Even when $w^*$ is known exactly, the size of the calibration set needed to achieve dataset-conditional guarantees will blow up as $r \to 0$. This happens because the density ratio blows up, allowing a few examples to wildly affect the score threshold. Additionally, when $w^*$ is unknown, and must be learned, the loss of coverage extends to expected guarantees, conditional on the dataset used for the density ratio estimation. This happens when $Q$ contains many examples in $\mathcal{B}$ but $P$ contains few or none — in this case, unconstrained density ratio estimation methods overestimate the density ratio on $\mathcal{B}$. We make these ideas formal in Appendix B.

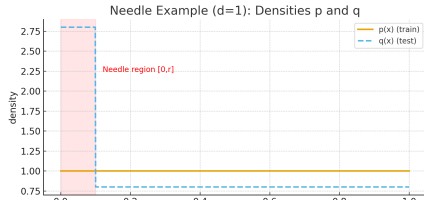

Figure 1: Visualization of Example 1. The red region contributes to the large second moment $\mathbb{E}_P[w^*(X)^2]$. By taking the width of this region to zero, we drive $\mathbb{E}_P[w^*(X)^2] \to \infty$.

Motivated by this example, we ask the following question: *Can we obtain reliable, dataset-conditional conformal coverage guarantees under covariate shift when the true density ratio is unbounded or must be learned from data?*

We answer this question in the affirmative. We propose a simple yet effective technique: clipping the class of density ratios. Instead of learning an unbounded density ratio, we propose clipped least-squares importance fitting (CLISF) to learn a ratio that is clipped at a threshold $B \geq 1$. This introduces a small, controllable bias but significantly reduces variance. By simply running WCP with $\hat{w}$ at a slightly inflated target coverage level, we restore expected and dataset-conditional coverage guarantees; we call this combined approach clipped weighted conformal prediction (CWCP).

Our main contributions are summarized below.

**A novel approach to stable density-ratio estimation.** We propose CLISF, which learns clipped density ratios $\hat{w} \in [0, B]$ via density ratio estimation on a *clipped* class, reducing variance and overfitting relative to unclipped estimators. This is a subtle but important distinction from the post-hoc clipping heuristic used by Tibshirani et al. (2019) and leads to provable generalization guarantees for the estimated density ratio. This is also distinct from methods which trim the *dataset* to exclude high variance points (Liu et al., 2017; Ma & Wang, 2020). Additionally, we show that we can accurately estimate the bias introduced by CLISF, which is necessary for downstream use in CP. To our knowledge, this is the first finite-sample theory for weight clipping in conformal inference.

**Finite-sample, dataset-conditional, two-sided guarantees.** We prove dataset-conditional coverage guarantees for CWCP with calibration size polynomial in $(B, \epsilon^{-1}, \log(1/\delta))$ and *no* dependence on higher moments of $w^*$. Furthermore, unlike much prior work which provides only *one-sided* coverage guarantees (Tibshirani et al., 2019; Joshi et al., 2025; Park et al., 2020) we provide stronger *two-sided* guarantees. Thus, CWCP *provably* does not achieve the target coverage guarantee by trivially overcovering. This result is a *dataset-conditional* analog to Proposition 1 of Lei & Candès (2021), under less restrictive assumptions on the higher moments of $w^*$. Our bounds represent a qualitative advancement for conformal prediction under realistic heavy-tailed shifts.

**Empirical validation.** We validate our algorithm on synthetic as well as real-world (iWildCam) datasets. Our method obtains tighter, more stable coverage than WCP under heavy tails.

## 1.1 RELATED WORK

**Reweighting methods for CP under covariate shift.** Importance weighting is a classical solution for covariate shift (Shimodaira, 2000). For conformal prediction, Tibshirani et al. (2019) proposed WCP. Subsequent work analyzed the case of learned weights, showing that coverage guarantees depend on the $L_1$-error of the weight estimate (Lei & Candès, 2021). However, obtaining good $L_1$ guarantees is challenging without assumptions like bounded density ratios or moments. This is also

the case for other density ratio-based methods (Park et al., 2021; Pournaderi & Xiang, 2024; Cortes et al., 2010; Joshi et al., 2025). Bhattacharyya & Barber (2024) assume *subpopulation structure* to estimate piecewise constant weights. Our work avoids such assumptions by clipping the weights.

**Alternative approaches to CP under distribution shift.** A parallel line of work calibrates predictors against a *distance* or *divergence* between $P$ and $Q$. Barber et al. (2023) prove that their NexCP algorithm has coverage bounded by a TV-like quantity measuring the shift between train and test points. Going beyond worst-case TV, Xu et al. (2025) bound the gap by the *Wasserstein-1* distance between the *score distributions* under $P$ and $Q$, yielding a tighter, shift-specific correction. Cauchois et al. (2024) and Ai & Ren (2024) use a distributionally robust optimization approach to guard against the worst-case shift in a ball centered at $P$.

**Variance reduction for importance weighting.** Sample trimming (Liu et al., 2017; Ma & Wang, 2020) is one approach to reduce the variance in importance weighting; this trims high-variance examples corresponding to a high estimated ratio. These methods focus on asymptotic consistency, whereas we are interested in a finite-sample $L_1$-error guarantee for the estimated ratio. Importance weight clipping is another approach (Ionides, 2008) and has been applied in CP as a heuristic to alleviate numerical issues with density ratio estimation (Tibshirani et al., 2019). In contrast to these methods, which apply clipping *post-hoc*, we integrate clipping directly in the density ratio estimation step, which leads to stronger guarantees when the weights must be learned.

## 2 PRELIMINARIES

**Density ratios and WCP.** Given distributions $Q \ll P$ over some space $\mathcal{Z}$, we define the density ratio (also known as the Radon-Nikodym derivative or importance weights) as $w^* = dQ/dP$. In this work, we will assume that $P$ and $Q$ admit density functions $p$ and $q$, so that $w^*(z) = q(z)/p(z)$ for all $z \in \mathcal{Z}$. If $\mathcal{Z} = \mathcal{X} \times \mathcal{Y}$ and $P, Q$ satisfy the covariate shift assumption, then $w^* = dQ/dP = dQ_X/dP_X$, that is, we can recover the importance weights between the $P$ and $Q$ from only their marginals $P_X$ and $Q_X$. The importance weights are useful because of the change of measure identity, $\mathbb{E}_P[w^*(Z) \cdot f(Z)] = \mathbb{E}_Q[f(Z)]$ for any measurable $f$, i.e., we can relate expectations under $Q$ to expectations under $P$. In particular, for a set of weights $w : \mathcal{X} \to \mathbb{R}_+$, we can define the weighted score CDF under a distribution $P$ over $\mathcal{X} \times \mathcal{Y}$ as

$$F_P(t, w) := \frac{\mathbb{E}_P[w(X) \cdot \mathbf{1}[s(X, Y) \le t]]}{\mathbb{E}_P[w(X)]} = \mathbb{E}_Q[\mathbf{1}[s(X, Y) \le t]] = F_Q(t), \tag{1}$$

where $Q$ is the distribution satisfying $dQ_X/dP_X = w/\mathbb{E}_P[w(X)]$. This is the motivation of WCP, which replaces the empirical CDF of nonconformity scores by a weighted empirical CDF

$$F_{(X_{\text{cal}}, Y_{\text{cal}})}(t, w) := \frac{\sum_{i=1}^m w(X_i) \cdot \mathbf{1}[s(X_i, Y_i) \le t]}{\sum_{i=1}^m w(X_i)}, \tag{2}$$

where $(X_{\text{cal}}, Y_{\text{cal}}) = (X_1, Y_1), \ldots, (X_m, Y_m)$ is a calibration set, and chooses the data–dependent cutoff $\tau := \inf \{t : F_m(t) \ge 1 - \alpha\} \cup \{\infty\}$.

**Density ratio estimation.** In practice, $w^*$ is not known exactly and must be learned. Prior work assumes access to some (typically parametric) class of density ratios $\mathcal{W} \subseteq \mathbb{R}^{\mathcal{X}}$ and aims to learn an approximate $\hat{w}$ using examples from $P_X$ and $Q_X$. A popular approach is least-squares importance fitting (LSIF) (Kanamori et al., 2009), which solves

$$\hat{w} = \arg\min_{w \in \mathcal{W}} \widehat{R}(w), \quad \text{where} \quad \widehat{R}(w) = \frac{1}{2m}\sum_{i=1}^m w(X_i)^2 - \frac{1}{n}\sum_{i=1}^n w(\tilde{X}_i). \tag{3}$$

Here, $X_P = (X_1, \ldots, X_m)$ and $X_Q = (\tilde{X}_1, \ldots, \tilde{X}_n)$ represent samples from $P_X$ and $Q_X$, respectively. Other popular approaches include KLIEP (Sugiyama et al., 2008), Kernel Mean Matching (Gretton et al., 2009), and source discriminators (Bickel et al., 2009). A key drawback, as mentioned earlier, is that (3) varies greatly over the draw of the sample when $\mathcal{W}$ is unbounded.

### 2.1 PROBLEM STATEMENT

Let $P$ and $Q$ be distributions over $\mathcal{X} \times \mathcal{Y}$ which are related by the covariate shift assumption. We access i.i.d. examples from $P$ and $Q_X$ via oracles $\text{EX}(P)$, $\text{EX}(P_X)$, and $\text{EX}(Q_X)$, from which we

may obtain some dataset $\mathcal{D}$. We do not assume access to $w^* = dQ/dP$. Instead, we assume access to some class of ratios $\mathcal{W}$, and assume $w^* \in \mathcal{W}$ (or a good approximation). Given $\alpha \in [0,1]$ and confidence $\delta \in [0,1]$ our goal is a threshold $\tau(\mathcal{D})$ satisfying the dataset-conditional guarantee

$$\Pr_{\mathcal{D}} \left[ \Pr_Q \left[ s(X,Y) \leq \tau(\mathcal{D}) \right] \geq 1 - \alpha \right] \geq 1 - \delta, \tag{4}$$

where $s : \mathcal{X} \times \mathcal{Y} \to \mathbb{R}$ is an arbitrary nonconformity score. Of course, this definition is too weak currently as it can easily be satisfied by outputting $\mathcal{Y}$ everywhere. Thus, we will also require upper bounds on the overcoverage, $\Pr_Q \left[ s(X,Y) \leq \tau(\mathcal{D}) \right] - (1 - \alpha)$. For our results, these upper bounds will be stated in terms of the bias $\Delta_B$ and an additional error parameter $\epsilon$.

## 3 CLIPPED LEAST-SQUARES IMPORTANCE FITTING

In this section, we formally introduce our algorithm for learning the clipped importance weights, which we call clipped least-squares importance fitting (CLISF). In place of the class $\mathcal{W}$, Algorithm 1 solves (3) over the *clipped* class

$$\mathcal{W}_B := \{x \mapsto \min(w(x), B) : w \in \mathcal{W}\}, \quad \text{for some } B \geq 1. \tag{5}$$

The advantage of this is two-fold. First, it requires fewer samples to obtain uniform convergence guarantees for (3) over $\mathcal{W}_B$ compared with $\mathcal{W}$. Second, using clipped weights leads to a more stable estimation of the population distribution of nonconformity scores when used downstream for WCP. This reduces the variance of the estimate of the $(1 - \alpha)$-coverage nonconformity score threshold.

Of course, by clipping the class $\mathcal{W}_B$, we introduce bias: if $\sup_{x \in \mathcal{X}} w^*(x) > B$, we will not be able to recover $w^*$ and so lose out on exact coverage guarantees. Assuming that $w^* \in \mathcal{W}$, We quantify this by the $L_1$-error between $w^*$ and the best approximation to $w^*$ in $\mathcal{W}_B$, the clipped true weights $w_B^*(x) = \min(w^*(x), B)$. For $B \geq 1$, this can be written as an $f$-divergence between $P$ and $Q$:

$$\Delta_B := \mathbb{E}_P[|w_B^*(X) - w^*(X)|] = \mathbb{E}_P[(w^*(X) - B)^+], \quad \text{where } x^+ := \max(x,0). \tag{6}$$

The clipping parameter $B$ allows us to toe this bias-variance tradeoff. When $B = 1$, then

$$\Delta_1 = \mathbb{E}_P[(w^*(X) - 1)^+] = \text{TV}(P, Q), \tag{7}$$

where TV is the total variation distance. When $B \geq \sup_{x \in \mathcal{X}} w^*(x)$, then clipping has no effect, so $\Delta_B = 0$ and we recover standard LSIF.

In the case that $w^* \notin \mathcal{W}$, we additionally present our results in terms of the misspecification:

$$\Delta_R = \inf_{w \in \mathcal{W}_B} R(w) - R(w_B^*), \quad \text{where } R(w) = \mathbb{E}_P[w(X)^2/2] - \mathbb{E}_Q[w(X)]. \tag{8}$$

Note that $w^* \in \mathcal{W}$ implies zero misspecification:

$$w^* \in \mathcal{W} \implies w_B^* \in \mathcal{W}_B \implies \Delta_R = 0. \tag{9}$$

---

**Algorithm 1** Clipped Least-Squares Importance Fitting (CLISF)

---

**Input:** Density ratios $\mathcal{W} \subseteq \mathbb{R}_+^{\mathcal{X}}$, coverage error $\epsilon \in (0,1]$, confidence $\delta \in (0,1]$, clipping parameter $B \in [1, \infty)$, example oracles $\text{EX}(P_X)$ and $\text{EX}(Q_X)$.
**Output:** Density ratio $\hat{w} : \mathcal{X} \to [0, B]$.
1: Set $X_{\text{train}} \leftarrow \text{EX}(P_X)^{m_{\text{train}}}$ and $X_{\text{test}} \leftarrow \text{EX}(Q_X)^{m_{\text{test}}}$, with $m_{\text{train}}$ and $m_{\text{test}}$ in Theorem 1.

2: Set $\hat{w} \leftarrow \arg\min_{w \in \mathcal{W}_B} \widehat{R}(w)$, with $\mathcal{W}_B$ as (5) and $\widehat{R}$ as (3).
3: Return $\hat{w}$.

---

Our analysis relies on a standard assumption in statistical learning theory. Our guarantees are presented for Rademacher classes of density ratios. Assumption 1 posits that we have access to a known upper bound on the complexity of $\mathcal{W}_B$. We additionally assume that the Rademacher complexity of the class decays at the standard $1/\sqrt{m}$ rate. This holds, for example, for linear classes (Shalev-Shwartz & Ben-David, 2014) and neural networks (Neyshabur et al., 2015).

**Assumption 1** (Bounded complexity of $\mathcal{W}_B$). *Let $X_{\text{train}} = (X_1, \ldots, X_m) \sim P^m$ and $X_{\text{test}} = (\tilde{X}_1, \ldots, \tilde{X}_n) \sim Q^n$. For any $B \in [1, \infty)$, we assume universal constants $C_B$ and $\tilde{C}_B$ such that*

$$\mathbb{E}_{X_{\text{train}}}[\text{Rad}_{X_{\text{train}}}(\mathcal{W}_B)] \leq C_B/\sqrt{m} \quad and \quad \mathbb{E}_{X_{\text{test}}}[\text{Rad}_{X_{\text{test}}}(\mathcal{W}_B)] \leq \tilde{C}_B/\sqrt{n}.$$

**Remark 1.** *Note that for any $B' > B$, $\mathcal{W}_B$ can be written as the composition of $\mathcal{W}_{B'}$ and the 1-Lipschitz clipping function $x \mapsto \min(x, B)$. Thus, it follows from Talagrand's contraction principle that $C_B$ and $\tilde{C}_B$ are nondecreasing in $B$, i.e., more clipping will always result in a larger reduction in the statistical complexity of the density ratio class.*[1]

Our analysis begins by establishing a connection between the LSIF objective function and the $L_2$-error of the learned weights with respect to the true clipped weights $w_B^*$. The following lemma provides an excess risk inequality tailored to our clipped setting. It shows that minimizing the population LSIF risk over the clipped class $\mathcal{W}_B$ is equivalent to finding the function in that class with the minimum squared error with respect to $w_B^*$.

**Lemma 1** (Excess risk transfer inequality for clipped ratios). *Let $P, Q$ and $w^* \in \mathcal{W}$ be as defined in Section 2.1. Define $\mathcal{W}_B$ as in (5). Define the clipped true weights $w_B^*(x) = \min(w^*(x), B)$. Then, it holds that $w_B^* \in \arg\min_{w \in \mathcal{W}_B} R(w)$. Additionally, for any $w \in \mathcal{W}_B$, it holds that*

$$\mathbb{E}_P\left[(w(X) - w_B^*(X))^2\right] \leq 2 \cdot (R(w) - R(w_B^*)).$$

We now provide a finite-sample generalization bound for the output of CLISF. By combining the result of Lemma 1 with standard uniform convergence guarantees for empirical risk minimization, the following theorem establishes that with high probability, CLISF returns a weight function $\hat{w}$ with low $L_2$-error relative to $w_B^*$ (the clipped true ratio). The sample complexity notably depends on the clipping parameter $B$ and the Rademacher complexity of the *clipped* function class $\mathcal{W}_B$.

**Theorem 1** ($L_2$-error generalization bound). *Assume Assumption 1 holds. Suppose we run Algorithm 1 with sample sizes $m_{\text{train}}, m_{\text{test}}$, where*

$$m_{\text{train}} = \mathcal{O}\left(\frac{B^2 C_B^2 + B^4 \log(1/\delta)}{\epsilon^2}\right), \quad m_{\text{test}} = \mathcal{O}\left(\frac{\tilde{C}_B^2 + B^2 \log(1/\delta)}{\epsilon^2}\right).$$

*Let $\hat{w}$ be the output of the call to Algorithm 1. Then, with probability at least $1 - \delta$,*

$$\mathbb{E}_P[(\hat{w}(X) - w_B^*(X))^2] \leq 2\Delta_R + \epsilon.$$

**Remark 2.** *When $P$ is known and $w^* \in \mathcal{W}$, we may without loss of generality remove any functions from $\mathcal{W}$ which integrate to more than 1 under $P$. This allows us to improve the dependence on $B$, which we formalize in Appendix A.3.*

**Remark 3.** *By Jensen's inequality, we can convert an $L_2$-error guarantee to an $L_1$-error guarantee,*

$$\mathbb{E}_P[(\hat{w}(X) - w_B^*(X))^2] \leq 2\Delta_R + \epsilon$$
$$\implies \mathbb{E}_P[|\hat{w}(X) - w_B^*(X)|] \leq \sqrt{2\Delta_R + \epsilon} \leq \sqrt{2\Delta_R} + \sqrt{\epsilon}.$$

**Remark 4.** *Theorem 1 does not say anything about the computational complexity of the clipped least-squares minimization problem. For example, for linear-in-features classes, for which the unclipped problem is convex (Kanamori et al., 2009), the clipped problem is nonconvex, and similar to ReLU regression, for which there are many hardness results (Goel et al., 2020). Thus, to guarantee Algorithm 1 is computationally efficient, we must make additional assumptions on $\mathcal{W}$.*[2]

## 3.1 ESTIMATING THE CLIPPING BIAS

The $B$-clipping bias defined in (6) can alternatively be written as

$$\Delta_B := \mathbb{E}_P[(w^*(X) - B)^+] = \mathbb{E}_P[w^*(X) - w_B^*(X)] = 1 - \mathbb{E}_P[w_B^*(X)]. \tag{10}$$

---

[1]See Appendix C for sharper bounds on the complexity of the clipped class, under additional assumptions.

[2]For example, it is sufficient to assume a piecewise constant structure as in Bhattacharyya & Barber (2024) or Park et al. (2021). We formalize this in Appendix D.

Motivated by the results above, suppose we have a clipped ratio estimate $\hat{w} : \mathcal{X} \to [0, B]$ such that $\mathbb{E}_P[|\hat{w}(X) - w_B^*(X)|] \leq \epsilon$. We define the bias estimate

$$\widehat{\Delta}_B := 1 - \frac{1}{m} \sum_{i=1}^{m} \hat{w}(X_i) \tag{11}$$

where $X_1, \ldots, X_m$ represent a *bias estimation* sample. Since $\hat{w}$ is bounded, we may apply concentration inequalities to show that $\widehat{\Delta}_B$ sharply concentrates around its expectation $1 - \mathbb{E}_P[\hat{w}(X)]$. Furthermore, $\mathbb{E}_P[\hat{w}(X)] \approx \mathbb{E}_P[w_B^*(X)]$ due to the $L_1$-error guarantee of $\hat{w}$. Thus, given a learned clipped ratio, we are may obtain a tight estimate of $\Delta_B$. This is summarized in the following lemma.

**Lemma 2.** *Suppose $\hat{w} : \mathcal{X} \to [0, B]$ satisfies $\mathbb{E}_P[|\hat{w}(X) - w_B^*(X)|] \leq \epsilon$. Let $X_1, \ldots, X_m \sim P_X$ be an i.i.d. sample. Then, for any $\gamma > 0$,*

$$\Pr\left[\left|\widehat{\Delta}_B - \Delta_B\right| > \epsilon + \gamma\right] \leq 2 \exp\left(-\frac{\gamma^2 m}{2B(1 + \epsilon + \gamma)}\right).$$

### 3.2 Choosing the Clipping Parameter

In this section, we discuss strategies to select the clipping parameter $B$. A good choice of $B$ is critical to balance the bias-variance tradeoff inherent in clipped importance weighting. A small $B$ aggressively clips the weights, which reduces the variance of the conformal predictor but introduces a potentially large clipping bias, leading to overcoverage. Conversely, a large $B$ reduces this bias but can lead to unstable predictors, especially when the true density ratio is unbounded.

Because the setting of $B$ affects the variance of the CLISF objective, conventional model selection techniques such as cross-validation can be unreliable. Cross-validation requires a stable estimate of out-of-sample performance to choose a hyperparameter. However, the CLISF objective itself can be a high-variance estimator, particularly for large values of $B$ that permit large weights. The objective contains a term quadratic in the weights, and when the true density ratio is heavy-tailed, this term makes the empirical risk highly sensitive to the specific data sample. As a result, the value of $B$ chosen by cross-validation can vary significantly with different random splits of the data.

**Choosing $B$ via structural risk minimization.** Structural risk minimization (SRM) (see Lugosi & Zeger (1996) and Koltchinskii (2001)) offers a data-driven approach for selecting $B$. Clipping $\mathcal{W}$ creates a hierarchy of increasingly complex function classes $\{\mathcal{W}_B : B \geq 1\}$. SRM selects the class from this hierarchy that minimizes an upper bound on the true risk. This involves choosing $B^*$ that minimizes the sum of the empirical CLISF risk minimizer and a complexity penalty derived from our uniform convergence bounds (Theorem 1), which depends on Rademacher complexity of $\mathcal{W}_{B^*}$. We empirically validate this approach in Appendix F.2.

**Other approaches.** See Appendix E for additional exploration of this topic.

## 4 Weighted Conformal Prediction with Clipped Weights

In this section, we analyze the performance of WCP when run with a clipped density ratio $\hat{w}$ learned by CLISF. We broadly refer to this approach as clipped weighted conformal prediction (CWCP).

### 4.1 Warmup: Expected Coverage Guarantees

As a warmup, we show that CWCP can restore the expected marginal coverage guarantee. The intuition is as follows: for a learned clipped density ratio $\hat{w}$ satisfying $\mathbb{E}_P[|\hat{w}(X) - w_B^*(X)|] \leq \epsilon$, the triangular inequality yields $\mathbb{E}_P[|\hat{w}(X) - w^*(X)|] \leq \Delta_B + \epsilon$. Thus, we can apply a similar result to Proposition 1 of Lei & Candès (2021) (see Lemma 9 in the appendix) to bound the expected undercoverage by $\Delta_B + \epsilon$. Since we can accurately estimate $\Delta_B$ (see Lemma 2), we can thus precisely estimate the correction we need to account for the error in the learned density ratio $\hat{w}$. Below, we make this intuition formal while also accounting for the misspecification $\Delta_R$.

**Theorem 2** (CWCP achieves expected coverage). *Suppose $\hat{w} : \mathcal{X} \to [0, B]$ satisfies $\mathbb{E}_P[|\hat{w}(X) - w_B^*(X)|] \leq \sqrt{2\Delta_R} + \epsilon$ and $\widehat{\Delta}_B \in \mathbb{R}$ satisfies $|\widehat{\Delta}_B - \Delta_B| \leq \sqrt{2\Delta_R} + 2\epsilon$. Suppose we run WCP*

*with weights $\hat{w}$ at a coverage level of $1 - \alpha + \widehat{\Delta}_B + 3\epsilon$, with an i.i.d. calibration set $X_{\text{cal}} = (X_1, Y_1), \ldots, (X_m, Y_m) \sim P$ and obtain prediction sets $C_\tau(x) = \{y \in Y : s(x, y) \leq \tau\}$. Then,*

$$1 - \alpha - 2\sqrt{2\Delta_R} \leq \Pr_{X_{\text{cal}}, Q}[Y \in C_\tau(X)].$$

To understand Theorem 2, let us first parse the conditions $\mathbb{E}_P[|\hat{w}(X) - w_B^*(X)|] \leq \sqrt{2\Delta_R} + \epsilon$ and $|\widehat{\Delta}_B - \Delta_B| \leq \sqrt{2\Delta_R} + 2\epsilon$. This separates the $L_1$-error of $\hat{w}$ and $\widehat{\Delta}_B$ into two components: a misspecification error $\sqrt{2\Delta_R}$, and a "finite-sample" error which must be $O(\epsilon)$. Note that, by combining Theorem 1, Remark 3, and Lemma 2, we may obtain $\hat{w}$ and $\widehat{\Delta}_B$ satisfying these conditions. By following this approach, note that we will know (an upper bound) on the finite-sample error (as the sample complexity bound of CLISF allows us to precisely control $\epsilon$ in terms of the sample size) but we *will not* be able to estimate $\Delta_R$. Thus, Theorem 2 states by slightly inflating the coverage by the term $\widehat{\Delta}_B + 3\epsilon$, we are able to correct the undercoverage due to the clipping bias $\Delta_B$ and the finite-sample error — in other words, the only source of undercoverage will be due to misspecification in the model class. This is to be expected: if there is misspecification in $\mathcal{W}$, then in general no algorithm can hope to exactly recover $w^*$ or $w_B^*$ in a reasonable number of samples.

## 4.2 DATASET-CONDITIONAL COVERAGE GUARANTEES

Next, we show that CWCP restores the dataset-conditional marginal coverage guarantee (4). Similar to the expected coverage setting, we run WCP with an inflated coverage level. Unlike Theorem 2, which holds regardless of the calibration set size, we now enforce that our calibration set is large enough to ensure that the weighted empirical CDF is a good approximation everywhere to the true distribution of nonconformity scores under $Q$. This relies on a weighted DKW inequality (see Pournaderi & Xiang (2024)), which is enabled by our use of clipped weights.

Our analysis relies on a standard assumption in conformal prediction for establishing upper bounds on coverage, that the CDF of the nonconformity scores is continuous. This is a mild technical condition that ensures quantiles are unique (see, e.g. Proposition 1 of Lei & Candès (2021) or Theorem 34 of Roth (2022)).

**Assumption 2.** *The cumulative distribution function of the nonconformity score is continuous.*

We additionally require that the true bias $\Delta_B$ is not too large. From (7), we know that $\Delta_B \leq 1$ for $B \geq 1$. We assume that $\Delta_B < 1$, i.e., that the bias is *strictly lower* than 1. Below, the choice of $1/2$ as the upper limit is arbitrary, and any choice in $(0, 1)$ will work with our proof, affecting only the final constants. Furthermore, since we control $B$, we may choose it large enough so that $\Delta_B \leq 1/2$ holds. Thus, we view this assumption as mild and primarily made for ease of exposition.

**Assumption 3.** *The bias is not too large: $\Delta_B \leq 1/2$.*

**Theorem 3** (CWCP achieves dataset-conditional coverage). *Assume Assumptions 2 and 3 hold. Suppose $\hat{w} : \mathcal{X} \to [0, B]$ satisfies $\mathbb{E}_P[|\hat{w}(X) - w_B^*(X)|] \leq \sqrt{2\Delta_R} + \epsilon$ and $\widehat{\Delta}_B \in \mathbb{R}$ satisfies $|\widehat{\Delta}_B - \Delta_B| \leq \sqrt{2\Delta_R} + 2\epsilon$, for some $\epsilon$ such that $\sqrt{2\Delta_R} + \epsilon \leq 1/4$. Suppose we run WCP with weights $\hat{w}$ at an inflated coverage level of $1 - \alpha + \widehat{\Delta}_B + 5\epsilon$, with an i.i.d. calibration set $X_{\text{cal}} = (X_1, Y_1), \ldots, (X_m, Y_m) \sim P$, where $m = \mathcal{O}\left(\frac{B\log(1/\epsilon) + B\log(1/\delta)}{\epsilon^2} + \frac{B^2\log(1/\delta)}{\epsilon^2}\right)$, and $C_\tau(x) = \{y \in \mathcal{Y} : s(x, y) \leq \tau\}$. Then,*

$$\Pr_{X_{\text{cal}}}\left[1 - \alpha - 2\sqrt{2\Delta_R} \leq \Pr_Q[Y \in C_\tau(X)] \leq 1 - \alpha + 2\Delta_B + 12\epsilon + 2\sqrt{2\Delta_R}\right] \geq 1 - \delta.$$

Theorem 3 requires that $\sqrt{2\Delta_R} + \epsilon \leq 1/4$ and thus requires that $\Delta_R < 1/32$ (note that we did not optimize the constant $1/32$ in this condition and it can likely be improved; however, we do not know how to remove the restriction that $\Delta_R = \mathcal{O}(1)$ from our proof). Nevertheless, we still believe this result to be of theoretical interest when $\mathcal{W}$ is sufficiently rich or structural assumptions are imposed on $P$ and $Q$ which inform the choice of a class $\mathcal{W}$ with zero misspecification error (for example, when $P$ and $Q$ are Gaussian, discrete, or piecewise constant, or when the density ratio is assumed to have a certain structure, such as linear-in-known-features).

By combining Theorem 3 with Theorem 1 and Lemma 2, we are able to obtain end-to-end high-probability dataset-conditional guarantees for CWCP with learned importance weights.

**Corollary 1** (End-to-end guarantees). *Assume that the conditions of Theorem 1, Lemma 2, and Theorem 3 hold. Suppose we first learn a clipped density ratio $\hat{w} : \mathcal{X} \to [0, B]$, where $\hat{w} \leftarrow \text{CLISF}(\mathcal{W}, \epsilon^2, \delta, B, \text{EX}(P_X), \text{EX}(Q_X))$ as in Theorem 1. Second, we use $\hat{w}$ as in Lemma 2 with an estimation sample size $m_{\text{est}} = \mathcal{O}(B \log(1/\delta)/\epsilon^2)$ to get a bias estimate $\widehat{\Delta}_B$. Third, we use $\hat{w}$ and $\Delta_B$ as in Theorem 3 to obtain prediction sets $C_\tau(X) = \{y \in \mathcal{Y} : s(x, y) \leq \tau\}$. Then,*

$$\Pr\left[1 - \alpha - 2\sqrt{2\Delta_R} \leq \Pr_Q[Y \in C_\tau(X)] \leq 1 - \alpha + 2\Delta_B + 12\epsilon - 2\sqrt{2\Delta_R}\right] \geq 1 - 3\delta$$

*where the randomness is over the draw of the density ratio estimation sets, the bias estimation set, and the calibration set. Additionally, we require*

$$\mathcal{O}\left(\frac{B\log(1/\epsilon) + B\log(1/\delta)}{\epsilon^2} + \frac{\log(1/\delta)}{\epsilon^2}\right), \mathcal{O}\left(\frac{B^2 C_B^2 + B^4 \log(1/\delta)}{\epsilon^4}\right), \mathcal{O}\left(\frac{\tilde{C}_B^2 + B^2 \log(1/\delta)}{\epsilon^4}\right)$$

*labeled examples from $P$, unlabeled examples from $P$, and unlabeled examples from $Q$, respectively.*

*Proof.* We union bound the failure events of Theorem 1, Lemma 2, and Theorem 3. □

### 4.3 Split Conformal vs. WCP vs. CWCP

At the end of the day, a practitioner might wonder when to use split conformal prediction, weighted conformal prediction, or clipped weighted conformal prediction. As evidenced by Example 1, CWCP is preferable to WCP when the true ratio $w^*$ has large higher moments under $P$ because it does not catastrophically undercover when the calibration set contains an input $x$ such that $w^*(x)$ is very large. However, a reader might note that, when applied to Example 1, split conformal will also perform well: since split conformal achieves $1 - \alpha$ expected marginal coverage on $P$, then it will also achieve at least $1 - \alpha - \text{TV}(P, Q)$ coverage on $Q$.

A few remarks are in order. First, under the setting of Theorem 3 shows that CWCP *does not* undercover (assuming no misspecification). In order to achieve the same guarantee with split conformal, a natural approach would be to inflate the prediction by a level of $\text{TV}(P, Q) = \Delta_1$ — this is the same correction used by CWCP with $B = 1$. In general, obtaining a good estimate of this quantity requires some machinery such as training a discriminative model (Sreekumar & Goldfeld (2022) and Tao et al. (2024)) or density ratio estimation. This remark is of particular interest when the resulting prediction sets are for downstream use by a risk-averse agent — in this case, it is important to ensure minimal undercoverage, as Theorem 3 does.

Second, a natural question is if there are problems which are (i) challenging for split conformal prediction, (ii) challenging for unclipped density ratio estimation methods and WCP, and (iii) not challenging for CWCP with a modest choice of $B$. In general, this will be the case when $w^*$ follows a power law. To illustrate this, let $P_X = U(0, 1)$ and define $w^*(x) = 1/(2\sqrt{x})$. It is easily checked that $w^*$ defines a valid density ratio with $\mathbb{E}_P[w^*(X)^2] = \infty$ (infinite second moment) and thus will present a challenge for (unclipped) LSIF and WCP. On the other hand, since the tail probability of $w^*$ is $P(w^*(X) \geq t) = 1/(4t^2)$, we have

$$\Delta_B = \mathbb{E}_P[(w^*(X) - B)^+] = \int_B^\infty P(w^*(X) \geq t) \, dt = \int_B^\infty \frac{1}{4t^2} \, dt = \frac{1}{4B}.$$

In particular, note that $\Delta_1 = 1/4$, which implies that split conformal will significantly undercover. On the other hand, by taking $B = \mathcal{O}(1/\epsilon)$, we may drive $\Delta_B \leq \epsilon$. In this example, CWCP is able to balance the advantages of both WCP (accounting for the covariate shift) and split conformal (low variance). In finance, power-law distributions are often used to model log-returns of a stock and trading volume (Gabaix et al., 2003); when training models on one time period (e.g., pre-crisis) and testing on another (e.g., during crisis), the density ratios between these periods naturally inherits this heavy-tailed behavior. In medical studies, extreme density ratios are also common (Li et al. (2019), Gao et al. (2021)). These lend credence to the practical applicability of CWCP.

## 5 Experiments

We compare our method with the following baselines: WCP + LSIF (Tibshirani et al., 2019) and likelihood-regularized quantile-regression (LR-QR) (Joshi et al., 2025). For illustration, to demon-

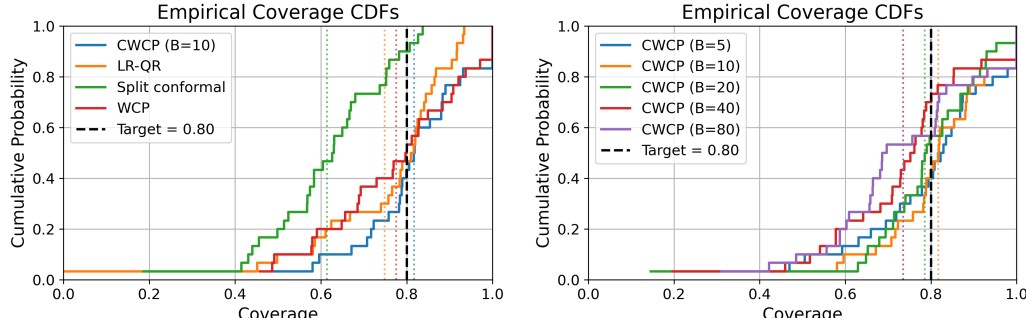

Figure 2: (Left) Coverage results for CWCP, LR-QR, split conformal, and WCP on iWildCam data. (Right) Ablation results for CWCP, varying $B$. The solid colored lines show the distribution of coverage levels over 30 trials. The colored dotted lines represent average coverage levels. Qualitatively, better performance is given by a CDF which looks like a step function about $0.8$.

strate the necessity of accounting for covariate shift, we additionally include split conformal (Papadopoulos et al., 2002) in our comparisons. For additional experiments, see Appendix F.

## 5.1 WILDLIFE CAMERA TRAP DATA

We evaluate our method on the *iWildCam* dataset (Beery et al., 2021), which contains 203029 datapoints corresponding to photographs taken by wildlife cameras across the globe. These images additionally have metadata containing a location identifier; this was only used to split the data. The task is to classify the image as one of $182$ species based on a $224 \times 224$ RGB photograph. The data was organized into $4$ splits: train, validation, in-distribution (ID) test, and out-of-distribution (OOD) test. No locations were shared between the {train, validation, ID test} and OOD test splits. Thus, a covariate shift arises from differences in camera choice, ambient light, etc.

**Experimental details.** The nonconformity score was $1 - p(x)$, where $p$ was a model trained beforehand on the train split to predict class probabilities. This was done by finetuning a linear head over the representation layer of a pretrained image model. $\mathcal{W}$ was defined similarly. To fit each conformal prediction method, we sample 20 locations from the train set and 20 locations from the OOD test set. We hold out half of the subsampled test set for evaluation; we discard the labels of the other half. We then train each method on the kept data and then find its coverage on the held out test set. We used a coverage level of $1 - \alpha = 0.8$. This was repeated for 30 trials.

**Results.** Figure 2 displays the results. Notably, split conformal has significant average undercoverage due to not accounting for covariate shift. WCP, CWCP, and LR-QR track the nominal coverage on average. Additionally, by inspecting the tails, we see that the coverage values of CWCP are the most tightly concentrated around the nominal value of $0.8$. We additionally performed an ablation study by varying $B$. Notably, for smaller values of the clipping parameter, the average coverage remained close to $0.8$. However, for $B = 40$ and $B = 80$, there was significant undercoverage.

## 5.2 SYNTHETIC DATA

We additionally evaluate our method on synthetic data at various controlled levels of covariate shift. The covariate is $X = (X_1, \dots, X_d) \in \mathbb{R}^d$ and the outcome is $Y \in \mathbb{R}$. We define

$$P_X := \mathcal{N}(0, I_d), \quad Q_X = \mathcal{N}(\beta \cdot e_1, I_d)$$

$$P_{Y|X} = Q_{Y|X} = \mathbf{1}^\top X + \exp(X_1^2) + \mathcal{N}(0, 1)$$

where $e_1$ is the first standard basis vector and $\beta$ models the level of covariate shift.

**Experimental details.** We consider the nonconformity score $s$ defined by the residual $|Y - \mu(X)|$, where $\mu : \mathbb{R}^d \to \mathbb{R}$ is a fixed regression model trained beforehand on $P$. We consider the class $\mathcal{W}$ of the general form of a change of measure between two Gaussians with identity covariance, $\mathcal{W} = \left\{ x \mapsto \exp\left( x^\top \mu - \frac{\|\mu\|^2}{2} \right) : \mu \in \mathbb{R}^d \right\}$. All algorithms are run with $d = 100$ with 600 examples.

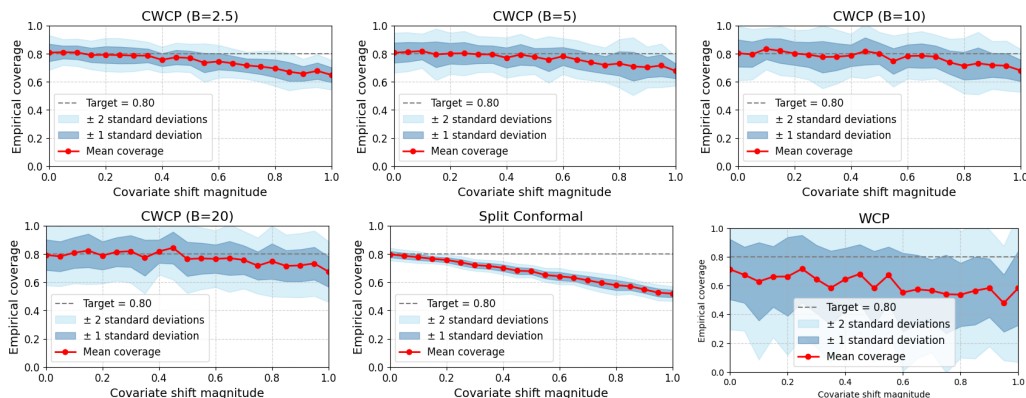

Figure 3: Coverage results for CWCP ($B \in \{2.5, 5, 10, 20\}$), split conformal, and WCP on synthetic shifted Gaussians data. The $x$-axis represents $\beta$. Qualitatively, good performance corresponds to a red line which is close to $y = 0.8$ (good expected coverage) and a small blue region (low variance).

For each value of $\beta \in [0, 0.1, 0.2, \ldots, 2]$, we ran 30 trials of the experiment above with a target of $1 - \alpha = 0.8$. For each trial, we measured the coverage on a freshly drawn dataset from $Q$.

**Results.** Figure 3 displays the results. We did not evaluate LR-QR, as $\mathcal{W}$ was not compatible with the linear structure assumed in Joshi et al. (2025). Notably, CWCP and split conformal had much less variance in coverage than WCP. However, split conformal displayed increasing levels of undercoverage with increasing $\beta$, where as this was less of an issue for CWCP and WCP (which account for the covariate shift). Comparing CWCP run with different levels of $B$, one can see that the variance increases as $B$ increases; however, for lower values of $B$ there was slight degradation of the expected coverage (this is most apparent when comparing $B = 2.5$ and $B = 20$).

## 6 CONCLUSION

We introduce a principled framework to address the instability of weighted conformal prediction under covariate shifts with unbounded density ratios. Our method consists of two components: CLISF, which learns stable density ratios by regularizing the function class, and CWCP, which constructs prediction sets and corrects for the clipping-induced bias. We provide dataset-conditional coverage guarantees for this approach. Crucially, the sample complexity of our method does not blow up with the higher moments of the density ratio, a key limitation of prior work. Experiments confirm that weight clipping is an effective tool for reliable conformal inference under shift.

To conclude, we outline possible directions for future work.

**Beyond marginal guarantees.** This work focuses on marginal coverage. An important next step is to extend this clipping-based framework to achieve stronger, more fine-grained guarantees, such as class-conditional or group-conditional coverage under covariate shift.

**Efficient alternatives to CLISF.** As mentioned in Remark 4, in general the CLISF problem is nonconvex. Future work could investigate convex surrogates or penalties instead of clipping.

**Correction only where necessary.** Our method adjusts for the clipping bias by inflating the coverage. Our overcoverage guarantee is thus averaged over the entire distribution $P$. However, one might hope for guarantees more akin to PQ-learning or learning with rejection (Goldwasser et al. (2020), Kalai & Kanade (2021)), in which the overcoverage should be limited to a specific subpopulation of $\mathcal{X}$. More formally, we would like to output a partition $\mathcal{X} = \mathcal{X}_1 \cup \mathcal{X}_2$, and achieve almost exact coverage conditioned on $X \in \mathcal{X}_1$, while guaranteeing that the mass of $\mathcal{X}_2$ under $P$ is small.

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

# A PROOFS

## A.1 PROBABILISTIC INEQUALITIES

**Lemma 3** (Bernstein's inequality)**.** *Let $Z_1, \ldots, Z_m$ be independent random variables such that $|Z_i - \mathbb{E}[Z_i]|$ are almost surely bounded by $M$. Let $\sigma^2 := \sum_{i=1}^m \operatorname{Var}(Z_i)$. Then, for all $t > 0$,*

$$\Pr\left[\left|\sum_{i=1}^m (Z_i - \mathbb{E}[Z_i])\right| > t\right] \leq 2\exp\left(-\frac{t^2}{2\sigma^2 + \frac{2}{3}Mt}\right).$$

**Lemma 4** (McDiarmid's inequality)**.** *Let $X_1, \ldots, X_m$ be independent random variables taking values in $\mathcal{X}$, and let $f : \mathcal{X}^m \to \mathbb{R}$ satisfy the* bounded differences property*: there exist constants $C_1, \ldots, C_m \geq 0$ such that for all $i$ and all $x_1, \ldots, x_m, x_i' \in \mathcal{X}$,*

$$|f(x_1, \ldots, x_i, \ldots, x_m) - f(x_1, \ldots, x_i', \ldots, x_m)| \leq C_i.$$

*Then, for any $\epsilon > 0$,*

$$\Pr\left[|f(X_1, \ldots, X_m) - \mathbb{E}[f(X_1, \ldots, X_m)]| \geq \epsilon\right] \leq 2\exp\left(-\frac{2\epsilon^2}{\sum_{i=1}^m C_i^2}\right).$$

**Lemma 5** (Bousquet's inequality, Bousquet (2002))**.** *Let $X_1, \ldots, X_n$ be independent identically distributed random vectors. Assume that $\mathbb{E}[X_{i,s}] = 0$, and that $X_{i,s} \leq 1$ for all $s \in \mathcal{T}$, where $\mathcal{T}$ is some index set. Let $v = 2\mathbb{E}[Z] + \sigma^2$ (where $\sigma^2 = \sup_{s \in \mathcal{T}} \sum_{i=1}^n \mathbb{E}[X_{i,s}^2]$). Then for all $t \geq 0$,*

$$\mathbb{P}\{Z \geq \mathbb{E}Z + t\} \leq \exp\left(-\frac{t^2}{2(v + t/3)}\right).$$

Below is a specialization of Lemma 5 to uniform convergence of a bounded function class.

**Lemma 6.** *Let $\mathcal{F}$ be a class of measurable functions taking values in $[0, B]$, and let $X_1, \ldots, X_n$ be i.i.d. random variables. Define $Z := \sup_{f \in \mathcal{F}} \left(\mathbb{E}[f(X)] - \frac{1}{n}\sum_{i=1}^n f(X_i)\right)$. Let $V := \sup_{f \in \mathcal{F}} \operatorname{Var}(f(X))$. Then for all $\delta \in (0, 1]$: with probability at least $1 - \delta$,*

$$Z \leq \mathbb{E}[Z] + \sqrt{\frac{2(V + 2\mathbb{E}[Z])\log(1/\delta)}{n}} + \frac{2B\log(1/\delta)}{3n}.$$

*Proof.* By applying Lemma 5 to the scaled random variables $f(X_i)/B$, we obtain

$$\Pr\left[Z > \mathbb{E}[Z] + \gamma\right] \leq \exp\left(-\frac{n\gamma^2}{2(V + 2\mathbb{E}[Z] + B\gamma/3)}\right).$$

We want this to be at most $\delta$. Solving for $\gamma$, it suffices for

$$\gamma \geq \sqrt{\frac{2(V + 2\mathbb{E}[Z])\log(1/\delta)}{n}} + \frac{2B\log(1/\delta)}{3n}.$$

$\square$

**Lemma 7** (Weighted DKW inequality, Pournaderi & Xiang (2024))**.** *Assume $Q \ll P$ and $w^* := dQ/dP \leq B$. Let $X_{\mathrm{cal}}, Y_{\mathrm{cal}} = (X_1, \ldots, X_m), (Y_1, \ldots, Y_m) \sim P^m$ be a calibration set. Denote by $F_{(X_{\mathrm{cal}}, Y_{\mathrm{cal}})}(t; w)$ and $F_P(t; w)$ the weighted empirical and population nonconformity score CDFs using $w$, respectively (see (1)). Then, for any $\gamma > 0$,*

$$\Pr_{X_{\mathrm{cal}}, Y_{\mathrm{cal}}}\left[\sup_{t \in \mathbb{R}} |F_{(X_{\mathrm{cal}}, Y_{\mathrm{cal}})}(t; w) - F_P(t; w)| > \gamma\right] \leq \frac{72}{\gamma}\exp\left(-\frac{m\gamma^2}{4B}\right) + 2\exp\left(-\frac{m\gamma^2}{2B^2}\right).$$

## A.2 PROOF OF LEMMA 1

Using the definition of the LSIF objective $R$ (Kanamori et al., 2009), we have

$$R(w) = \frac{1}{2} \cdot \mathbb{E}_P[(w(X) - w^*(X))^2] + C, \quad \forall w : \mathcal{X} \to \mathbb{R}_+,$$

where $C$ is some constant independent of $w$ and $w^* = dQ_X/dP_X$ is the true density ratio.

To prove the first claim, note that $w_B^* = \min(w^*(x), B)$ is the pointwise minimizer to the squared deviation $(w(x) - w^*(x))^2$ over $w \in \mathcal{W}_B$ for all $x \in \mathcal{X}$. This is due to the following two cases: if $w^*(x)$ (the unconstrained minimizer) lies in $[0, B]$, then we have $w_B^*(x) = w^*(x)$; alternatively, if $w^*(x) > B$, then the constrained minimizer is $B = w_B^*(x)$.

To prove the second claim, note that for any $w \in \mathcal{W}_B$,

$$
(w(x) - w^*(x))^2 - (w_B^*(x) - w^*(x))^2 = \begin{cases} (w(x) - w^*(x))^2, & w^*(x) \le B \\ (w(x) - w^*(x))^2 - (B - w^*(x))^2, & w^*(x) > B \end{cases}
$$

$$
= \begin{cases} (w(x) - w^*(x))^2, & w^*(x) \le B \\ (w(x) - B)(w(x) + B - 2w^*(x)), & w^*(x) > B \end{cases}
$$

$$
\ge \begin{cases} (w(x) - w^*(x))^2, & w^*(x) \le B \\ (w(x) - B)^2, & w^*(x) > B \end{cases}
$$

$$
= (w(x) - w_B^*(x))^2, \quad \forall x \in \mathcal{X}
$$

where the second equality is due to difference of squares and the third inequality is due to the case $w^*(x) > B$ combined with the fact that $w(x) \le B$ (since we assume $w \in \mathcal{W}_B$). Thus, integrating this entire inequality with respect to $P_X$, we obtain

$$
\mathbb{E}_P\big[(w(X) - w_B^*(X))^2\big] \le \mathbb{E}_P\big[(w(x) - w^*(x))^2 - (w_B^*(x) - w^*(x))^2\big]
$$
$$
= 2 \cdot (R(w) - R(w_B^*))
$$

which concludes the proof.

## A.3 PROOF OF THEOREM 1

First, we state and prove a supporting lemma. Below, we parameterize the upper bound on the expectation of functions from $\mathcal{W}_B$ by $U$, rather than a coarse bound by $B$, to account for scenarios where this upper bound might in fact be much less than $B$. For example, since $\mathbb{E}_P[w^*(X)] = 1$, one might expect that $U \ll B$ when $\mathcal{W}$ contains only those functions close to $w^*$.

**Lemma 8** (Uniform convergence of LSIF loss over bounded ratio class). *Suppose $\mathcal{W}_B \subseteq [0, B]^{\mathcal{X}}$ and $\mathbb{E}_P[w(X)] \le U$ for all $w \in \mathcal{W}_B$. Let $X_{\text{train}} = (X_1, \ldots, X_{m_{\text{train}}}) \sim P_X^{m_{\text{train}}}$ and $X_{\text{test}} = (\tilde{X}_1, \ldots, \tilde{X}_{m_{\text{test}}}) \sim Q_X^{m_{\text{test}}}$ be i.i.d. samples. Then for any $\delta \in (0, 1)$, with probability at least $1 - \delta$ over the draw of $X_{\text{train}}, X_{\text{test}}$,*

$$
\sup_{w \in \mathcal{W}} \left| \widehat{R}(w) - R(w) \right| \tag{12}
$$

$$
\le 2B \cdot \mathbb{E}_{X_{\text{train}}}[\text{Rad}_{X_{\text{train}}}(\mathcal{W})] + \sqrt{\frac{10UB^3 \log(2/\delta)}{m_{\text{train}}}} + \frac{B^2 \log(2/\delta)}{3m_{\text{train}}}
$$

$$
+ 2 \cdot \mathbb{E}_{X_{\text{test}}}[\text{Rad}_{X_{\text{test}}}(\mathcal{W})] + \sqrt{\frac{6UB \log(2/\delta)}{m_{\text{test}}}} + \frac{2B \log(2/\delta)}{3m_{\text{test}}}.
$$

*Proof.* By the triangular inequality,

$$
(12) \le \underbrace{\sup_{w \in \mathcal{W}} \left| \frac{1}{m_{\text{train}}} \sum_{i=1}^{m_{\text{train}}} \frac{w(X_i)^2}{2} - \mathbb{E}_{P_X}\left[\frac{w(X)^2}{2}\right] \right|}_{(A)} + \underbrace{\sup_{w \in \mathcal{W}} \left| \frac{1}{m_{\text{test}}} \sum_{i=1}^{m_{\text{test}}} w(\tilde{X}_i) - \mathbb{E}_{Q_X}[w(\tilde{X})] \right|}_{(B)}.
$$

We control (A) and (B) separately via uniform convergence arguments.

**Term (A).** Let $\mathcal{F}_{\text{train}} = \{x \mapsto \frac{1}{2}w(x)^2 : w \in \mathcal{W}\}$. Since $w(x) \in [0, B]$, each $f \in \mathcal{F}_{\text{train}}$ takes values in $[0, B^2/2]$, so $\text{Range}(\mathcal{F}_{\text{train}}) = B^2/2$. By Lemma 6,

$$(\text{A}) \leq \mathbb{E}_{X_{\text{train}}}[(\text{A})] + \sqrt{\frac{2(\sup_{f \in \mathcal{F}_{\text{train}}}(\text{Var}_P[f(X)]) + 2\mathbb{E}_{X_{\text{train}}}[(\text{A})])\log(1/\delta_{\text{train}})}{m_{\text{train}}}} + \frac{2(B^2/2)\log(1/\delta_{\text{train}})}{3m_{\text{train}}}$$

with probability at least $1 - \delta_{\text{train}}$. By a standard symmetrization argument,

$$\mathbb{E}_{X_{\text{train}}}[(\text{A})] \leq 2 \cdot \mathbb{E}_{X_{\text{train}}}[\text{Rad}_{X_{\text{train}}}(\mathcal{F}_{\text{train}})] \leq 2B \cdot \mathbb{E}_{X_{\text{train}}}[\text{Rad}_{X_{\text{train}}}(\mathcal{W})]$$

where the last inequality follows from the composition principle: since $\mathcal{F}_{\text{train}} = (r \mapsto r^2/2) \circ \mathcal{W}$, and since $r \mapsto r^2/2$ is $B$-Lipschitz for $r \in [0, B]$. Note that we also have the loose bound $2B \cdot \mathbb{E}_{X_{\text{train}}}[\text{Rad}_{X_{\text{train}}}(\mathcal{W})] \leq 2B^2$ since $\mathcal{W}$ is bounded by $B$. Next, note that

$$\sup_{f \in \mathcal{F}_{\text{train}}} \text{Var}_P[f(X)] \leq \sup_{f \in \mathcal{F}_{\text{train}}} \mathbb{E}_P[f(X)^2] = \sup_{w \in \mathcal{W}} \mathbb{E}_P[w(X)^4/4] \leq UB^3/4$$

where the last inequality follows from the assumption that $\mathbb{E}_P[w(X)] \leq U$ and $w(X) \leq B$ for all $w \in \mathcal{W}$. By combining these bounds, we find

$$(\text{A}) \leq 2B \cdot \mathbb{E}_{X_{\text{train}}}[\text{Rad}_{X_{\text{train}}}(\mathcal{W})] + \sqrt{\frac{10UB^3\log(1/\delta_{\text{train}})}{m_{\text{train}}}} + \frac{B^2\log(1/\delta_{\text{train}})}{3m_{\text{train}}}.$$

**Term (B).** Again by Lemma 6,

$$(\text{B}) \leq \mathbb{E}_{X_{\text{test}}}[(\text{B})] + \sqrt{\frac{2(\sup_{w \in \mathcal{W}}(\text{Var}_P[w(X)]) + 2\mathbb{E}_{X_{\text{test}}}[(\text{B})])\log(1/\delta_{\text{test}})}{m_{\text{test}}}} + \frac{2B\log(1/\delta_{\text{test}})}{3m_{\text{test}}}$$

$$\leq 2 \cdot \mathbb{E}_{X_{\text{test}}}[\text{Rad}_{X_{\text{test}}}(\mathcal{W})] + \sqrt{\frac{6UB\log(1/\delta_{\text{test}})}{m_{\text{test}}}} + \frac{2B\log(1/\delta_{\text{test}})}{3m_{\text{test}}}.$$

where the last line follows by bounded $\sup_{w \in \mathcal{W}}(\text{Var}_P[w(X)]) \leq UB$, again using the assumption that $\mathbb{E}_P[w(X)] \leq U$ and $w(X) \leq B$ for all $w \in \mathcal{W}$; and the coarse bound $\mathbb{E}_{X_{\text{test}}}[(\text{B})] \leq B$.

The desired result follows by combining our bounds on (A) and (B) together with a union bound, after choosing $\delta_{\text{train}} = \delta_{\text{test}} = \delta/2$. $\qquad\qquad\square$

We are now ready to give the proof of Theorem 1.

**Controlling the empirical process.** First, note that Lemma 8 holds with $U \leq B$ because we assume $\mathcal{W}_B \subseteq [0, B]^{\mathcal{X}}$. Thus, by Lemma 8 and Assumption 1,

$$\sup_{w \in \mathcal{W}_B} \left| \widehat{R}(w) - R(w) \right|$$

$$\leq 2B \cdot \mathbb{E}_{X_{\text{train}}}[\text{Rad}_{X_{\text{train}}}(\mathcal{W}_B)] + \sqrt{\frac{10B^4\log(2/\delta)}{m_{\text{train}}}} + \frac{B^2\log(2/\delta)}{3m_{\text{train}}}$$

$$+ 2 \cdot \mathbb{E}_{X_{\text{test}}}[\text{Rad}_{X_{\text{test}}}(\mathcal{W}_B)] + \sqrt{\frac{6B^2\log(2/\delta)}{m_{\text{test}}}} + \frac{2B\log(2/\delta)}{3m_{\text{test}}}$$

$$\leq \frac{2BC_B}{\sqrt{m_{\text{train}}}} + \sqrt{\frac{10B^4\log(2/\delta)}{m_{\text{train}}}} + \frac{B^2\log(2/\delta)}{3m_{\text{train}}} + \frac{2\tilde{C}_B}{\sqrt{m_{\text{test}}}} + \sqrt{\frac{6B^2\log(2/\delta)}{m_{\text{test}}}} + \frac{2B\log(2/\delta)}{3m_{\text{test}}}$$

with probability at least $1 - \delta$ over the draw of $X_{\text{train}}, X_{\text{test}}$. Shortly, we will require that the right hand of this is at most $\epsilon/4$. By making each term no more than $\epsilon/24$, this is the case if

$$m_{\text{train}} \geq \max\left(\frac{2304B^2C_B^2}{\epsilon^2}, \frac{5760B^4\log(2/\delta)}{\epsilon^2}, \frac{8B^2\log(2/\delta)}{\epsilon}\right) = \mathcal{O}\left(\frac{B^2C_B^2 + B^4\log(1/\delta)}{\epsilon^2}\right),$$

$$m_{\text{test}} \geq \max\left(\frac{2304\tilde{C}_B^2}{\epsilon^2}, \frac{3456B^2\log(2/\delta)}{\epsilon^2}, \frac{16B\log(2/\delta)}{\epsilon}\right) = \mathcal{O}\left(\frac{\tilde{C}_B^2 + B^2\log(1/\delta)}{\epsilon^2}\right).$$

**Applying the excess risk transfer lemma.** If the right hand above is bounded by $\epsilon/4$, then since $\hat{w}$ minimizes the empirical CLISF objective,

$$
\begin{aligned}
R(\hat{w}) &\leq \inf_{w \in \mathcal{W}_B} R(w) + 2 \cdot \epsilon/4 \\
&= R(w_B^*) + \inf_{w_B \in \mathcal{W}_B} (R(w_B) - R(w_B^*)) + \epsilon/2 \\
&= R(w_B^*) + \Delta_R + \epsilon/2 \\
&\implies \mathbb{E}_P[(\hat{w}_B(X) - w_B^*(X))^2] \leq 2\Delta_R + \epsilon
\end{aligned}
$$

where the last implication follows from Lemma 1. This concludes the proof.

**Remark 5.** *As mentioned in the statement of Theorem 1, we can improve the sample dependence on $B$ when $P$ is known. In this case, we need only consider functions which integrate to $1$, which represent the valid density ratios. In this case, it suffices to have*

$$
m_{\text{train}} = \mathcal{O}\left(\frac{B^2 C_B^2 + B^3 \log(1/\delta)}{\epsilon^2}\right), \quad m_{\text{test}} = \mathcal{O}\left(\frac{\tilde{C}_B^2 + B \log(1/\delta)}{\epsilon^2}\right).
$$

### A.4 PROOF OF LEMMA 2

By Bernstein's inequality,

$$
\Pr\left[\left|\widehat{\Delta}_B - (1 - \mathbb{E}_P[\hat{w}(X)])\right| > \gamma\right] = \Pr\left[\left|\frac{1}{m}\sum_{i=1}^m \hat{w}(X_i) - \mathbb{E}_P[\hat{w}(X)]\right| > \gamma\right]
$$
$$
\leq 2\exp\left(-\frac{\gamma^2 m^2}{2m \cdot \text{Var}_P[\hat{w}(X)] + \frac{2}{3}B\gamma m}\right). \tag{13}
$$

Next, note that

$$
\begin{aligned}
\text{Var}_P[\hat{w}(X)] &\leq \mathbb{E}_P[\hat{w}(X)^2] &&(\hat{w} \text{ is nonnegative}) \\
&\leq B \cdot \mathbb{E}_P[\hat{w}(X)] &&(\hat{w} \text{ is bounded above by } B) \\
&\leq B \cdot (\mathbb{E}_P[w_B^*(X)] + \epsilon) &&(\text{triangular inequality and } \mathbb{E}_P[|\hat{w}(X) - w_B^*(X)|] \leq \epsilon) \\
&\leq B \cdot (\mathbb{E}_P[w^*(X)] + \epsilon) = (1 + \epsilon)B. &&(w_B^* \leq w^* \text{ and } w^* \text{ integrates to } 1)
\end{aligned}
$$

Thus, plugging into (13) and performing some slight simplifications,

$$
(13) \leq 2\exp\left(-\frac{\gamma^2 m^2}{2m \cdot B(1+\epsilon) + \frac{2}{3}B\gamma m}\right) \leq 2\exp\left(-\frac{\gamma^2 m}{2B(1+\epsilon+\gamma)}\right),
$$

Finally, by several applications of the triangular inequality

$$
\begin{aligned}
&\left|\widehat{\Delta}_B - (1 - \mathbb{E}_P[\hat{w}(X)])\right| \leq \gamma \\
&\implies \left|\widehat{\Delta}_B - \Delta_B\right| \leq \left|\widehat{\Delta}_B - (1 - \mathbb{E}_P[\hat{w}(X)])\right| + |\mathbb{E}_P[\hat{w}(X)] - \mathbb{E}_P[w_B^*(X)]| \leq \epsilon + \gamma
\end{aligned}
$$

which concludes the proof.

### A.5 PROOF OF THEOREM 2

First, we state and prove a supporting lemma. This is a generalization of Proposition 1 of Lei & Candès (2021) to account for weights $w_1$ and $w_2$ which are not necessarily normalized to $1$. This arises due to weight clipping in Algorithm 1.

**Lemma 9.** *Let $P, Q, \mathcal{W}$ be as in Lemma 8. Let $w_1, w_2 \in \mathcal{W}$. Then,*

$$
\sup_{t \in \mathbb{R}} \left|F_P(t, w_1) - F_P(t, w_2)\right| \leq \frac{\mathbb{E}_P\left[|w_1(X) - w_2(X)|\right]}{\max\left(\mathbb{E}_P\left[w_1(X)\right], \mathbb{E}_P\left[w_2(X)\right]\right)}.
$$

*where $F_P(t, w)$ denotes the weighted nonconformity score CDF defined in (1).*

*Proof.* Write $C := \mathbb{E}_P\big[w_1(X)\big]$ and $D := \mathbb{E}_P\big[w_2(X)\big]$. Let $Q_1$ be the measure satisfying $d(Q_1)_X/dP_X = w_1/C$, and define $Q_2$ analogously for $w_2/D$. Then,

$$\sup_{t \in \mathbb{R}} \left| F_P(t, w_1) - F_P(t, w_2) \right| \leq \mathrm{TV}(Q_1, Q_2)$$

$$= \frac{1}{2} \mathbb{E}_P[|w_1(X)/C - w_2(X)/D|]$$

$$= \frac{1}{2} \mathbb{E}_P\left[ \left| \frac{w_1(X) - w_2(X)}{C} + (1/C - 1/D)w_2(X) \right| \right]$$

$$\leq \frac{\mathbb{E}_P[|w_1(X) - w_2(X)|]}{2C} + \frac{|D - C|}{2C}$$

$$\leq 2 \cdot \frac{\mathbb{E}_P[|w_1(X) - w_2(X)|]}{2C}$$

where the last line follows from the triangular inequality. Note that this argument is completely symmetric in $w_1$ and $w_2$, and so we may replace $C$ with $\max(C, D)$. This concludes the proof. □

We are now ready to give the proof of Theorem 2.

Our starting point is Corollary 1 of Tibshirani et al. (2019), which implies that

$$1 - \alpha + \widehat{\Delta}_B + 3\epsilon \leq \Pr_{X_{\mathrm{cal}}, (X,Y) \sim \widehat{Q}}[Y \in C_\tau(X)] = F_P(\tau, \hat{w})$$

where $\widehat{Q}$ is the measure satisfying $d\widehat{Q}/dP = \hat{w}/\mathbb{E}_P[\hat{w}(X)]$. Thus, by Lemma 9,

$$|F_P(\tau, \hat{w}) - F_P(\tau, w^*)| \leq \frac{\mathbb{E}_P[|\hat{w}(X) - w^*(X)|]}{1} \qquad (w^* \text{ integrates to } 1)$$

$$\leq \mathbb{E}_P[|\hat{w}(X) - w_B^*(X)|] + \mathbb{E}_P[|\hat{w}_B^*(X) - w^*(X)|]$$

$$\leq \epsilon + \sqrt{2\Delta_R} + \Delta_B$$

$$\leq 2\sqrt{\Delta_R} + \widehat{\Delta}_B + 3\epsilon \qquad (\text{we assume } |\widehat{\Delta}_B - \Delta_B| \leq 2\epsilon + \sqrt{2\Delta_R})$$

which implies that

$$F_P(\tau, w^*) = \Pr_{X_{\mathrm{cal}}, Q}[Y \in C_\tau(X)]$$

$$\geq 1 - \alpha + \widehat{\Delta}_B + 3\epsilon - (\widehat{\Delta}_B + 3\epsilon + 2\sqrt{2\Delta_R}) = 1 - \alpha - 2\sqrt{2\Delta_R}$$

which concludes the proof.

## A.6 PROOF OF THEOREM 3

We start by applying Lemma 7 to the normalized weights $\hat{w}/\mathbb{E}_P[\hat{w}(X)]$,

$$\Pr_{X_{\mathrm{cal}}, Y_{\mathrm{cal}}} \left[ \sup_{t \in \mathbb{R}} \left| F_{(X_{\mathrm{cal}}, Y_{\mathrm{cal}})}(t; \hat{w}_B) - F_P(t; \hat{w}_B) \right| > \epsilon \right]$$

$$\leq \frac{72}{\epsilon} \exp\left( -\frac{m\epsilon^2}{4(B/\mu)} \right) + 2 \exp\left( -\frac{m\epsilon^2}{2(B/\mu)^2} \right) \qquad (14)$$

where that $\mu = \mathbb{E}_{P_X}[\hat{w}_B(X)]$. We can lower bound $\mu$ as

$$\mu = \mathbb{E}_P[\hat{w}(X)]$$

$$\geq \mathbb{E}_P[w_B^*(X)] - \mathbb{E}_P[|\hat{w}(X) - w_B^*(X)|] \qquad (\text{triangular inequality})$$

$$\geq \mathbb{E}_P[w^*(X)] - \mathbb{E}_P[|w_B^*(X) - w^*(X)|] - \mathbb{E}_P[|\hat{w}(X) - w_B^*(X)|] \qquad (\text{triangular inequality})$$

$$\geq 1 - \Delta_B - (\sqrt{2\Delta_R} + \epsilon)$$

$$\qquad\qquad (\mathbb{E}_P[|w_B^*(X) - w^*(X)|] = \Delta_B \text{ and } \mathbb{E}_P[|\hat{w}(X) - w_B^*(X)|] \leq \sqrt{2\Delta_R} + \epsilon)$$

$$\geq 1/4. \qquad (\text{Assumption 3 and } \sqrt{2\Delta_R} + \epsilon \leq 1/4)$$

Substituting this lower bound into (14) gives the bound

$$(14) \leq \frac{72}{\epsilon} \exp\left(-\frac{m\epsilon^2}{16B}\right) + 2\exp\left(-\frac{m\epsilon^2}{32B^2}\right). \tag{15}$$

To ensure that (15) is at most $\delta$, it suffices to choose

$$m \geq \max\left(\frac{16B}{\epsilon^2}\log\left(\frac{144}{\epsilon\delta}\right), \frac{32B^2\log(4/\delta)}{\epsilon^2}\right).$$

Let this success event be denoted by $\mathcal{E}$. Casing on $\mathcal{E}$, we have

$$\sup_{t\in\mathbb{R}} \left| F_{(X_{\mathrm{cal}}, Y_{\mathrm{cal}})}(t, \hat{w}) - F_P(t, w^*) \right|$$

$$\leq \sup_{t\in\mathbb{R}} \left| F_{(X_{\mathrm{cal}}, Y_{\mathrm{cal}})}(t, \hat{w}) - F_P(t, \hat{w}) \right| + \sup_{t\in\mathbb{R}} \left| F_P(t, \hat{w}) - F_P(t, w_B^*) \right| + \sup_{t\in\mathbb{R}} \left| F_P(t, w_B^*) - F_P(t, w^*) \right|$$
$$\text{(triangular inequality)}$$

$$\leq \epsilon + \frac{\mathbb{E}_P[|\hat{w} - w_B^*|]}{\max(\mathbb{E}_P[\hat{w}(X)], \mathbb{E}_P[w_B^*(X)])} + \frac{\mathbb{E}_P[|w_B^*(X) - w^*(X)|]}{\mathbb{E}_P[w^*(X)]} \qquad (\mathcal{E} \text{ and Lemma 9})$$

$$\leq \epsilon + 2(\epsilon + \sqrt{2\Delta_R}) + \mathbb{E}_P[|w_B^*(X) - w^*(X)|]$$
$$(w^* \text{ integrates to 1, } \Delta_B \leq 1/2, \text{ and } \mathbb{E}_P[|w_B^*(X) - w^*(X)|] \leq \epsilon + \sqrt{2\Delta_R})$$

$$= \Delta_B + 3\epsilon + 2\sqrt{2\Delta_R} \tag{16}$$

Next, recall that WCP will output the score threshold

$$\tau := \inf\{t \in \mathbb{R} : F_{(X_{\mathrm{cal}}, Y_{\mathrm{cal}})}(t, \hat{w}) \geq 1 - \alpha + \widehat{\Delta}_B + 5\epsilon\}.$$

Note that since $F_{(X_{\mathrm{cal}}, Y_{\mathrm{cal}})}(t, \hat{w})$ is not continuous, it is not necessarily true that $F_{(X_{\mathrm{cal}}, Y_{\mathrm{cal}})}(\tau, \hat{w}) = 1 - \alpha + \widehat{\Delta}_B + 5\epsilon$. However, we show that the discretization error cannot be too large: casing on $\mathcal{E}$, and using Assumption 2, it holds that

$$1 - \alpha + \widehat{\Delta}_B + 5\epsilon \leq F_{(X_{\mathrm{cal}}, Y_{\mathrm{cal}})}(\tau, \hat{w}) \leq 1 - \alpha + \widehat{\Delta}_B + 7\epsilon. \tag{17}$$

(where we have used the continuity of $F_P(t)$ (which implies continuity of $F_P(t, \hat{w})$) in conjunction with the uniform error bound of $\mathcal{E}$ to argue that the "jumps" can be no more than $2\epsilon$). Thus,

$$1 - \alpha + \widehat{\Delta}_B + 5\epsilon \leq F_{(X_{\mathrm{cal}}, Y_{\mathrm{cal}})}(\tau, \hat{w}) \leq 1 - \alpha + \widehat{\Delta}_B + 7\epsilon$$

$$\implies 1 - \alpha + \Delta_B + 3\epsilon \leq F_{(X_{\mathrm{cal}}, Y_{\mathrm{cal}})}(\tau, \hat{w}) \leq 1 - \alpha + \Delta_B + 9\epsilon \qquad (|\widehat{\Delta}_B - \Delta_B| \leq 2\epsilon)$$

$$\implies 1 - \alpha - 2\sqrt{2\Delta_R} \leq F_P(\tau, w^*) \leq 1 - \alpha + 2\Delta_B + 12\epsilon + 2\sqrt{2\Delta_R} \qquad \text{(using (16))}$$

This concludes the proof, since $F_P(\tau, w^*) = Q(Y \in C_\tau(X))$.

# B  MOTIVATING EXAMPLE

For convenience, we restate Example 1 from the introduction.

**Example 1** (Restatement). Fix a dimension $d \in \mathbb{N}$, radius $r \in (0, 1)$, and mixture weight $\theta \in (0, 1)$. Define the input space $\mathcal{X} = [0, 1]^d$ and label space $\mathcal{Y} = [0, 1]$. Define $\mathcal{B}$ to be the ball $\{x \in \mathcal{X} : \|x\|_\infty \leq r\}$. Define the train distribution $P$ to be uniform over $\mathcal{X} \times \mathcal{Y}$. Define the test distribution $Q = (1-\theta)P + \theta S$, where $S$ is uniform over $\mathcal{B} \times \mathcal{Y}$. Define the nonconformity score to be $s(x, y) = \|x\|_\infty$. It can be checked that $\mathrm{TV}(P, Q) = \theta(1 - r^d)$ and $w^*(x) = \begin{cases} 1 - \theta + \theta/r^d, & x \in \mathcal{B} \\ 1 - \theta, & x \notin \mathcal{B} \end{cases}$.

In this example, as $r \to 0$, note that $\mathrm{TV}(P, Q) \to \theta$ but $\sup_{x\in\mathcal{X}} w(x) \to \infty$. In other words, as the radius decreases, the total variation between $P$ and $Q$ remains stable, but the supremum of the density ratio is unbounded.

**Proposition 1.** *Fix parameters $d \in \mathbb{N}, r \in (0,1), \theta \in (0,1), \alpha \in (0,1)$ with $\theta < 1 - \alpha$. Let distributions $P, Q$, ball $\mathcal{B}$, true density ratio $w^*$, and score $s$ be as in Example 1. Suppose*

$$m = \left\lfloor \frac{c}{r^d} \right\rfloor, \quad where \ 0 < c < \frac{\alpha\theta}{(1-\alpha)(1-\theta)}. \tag{18}$$

*Suppose we draw the calibration set $X_{\mathrm{cal}} = (X_1, \dots, X_m) \sim P^m$ and compute the WCP threshold $\tau$ using the true density ratio $w^*$. Then, with probability at least $1 - e^{-(c-r^d)}$, the score threshold satisfies $\tau \leq r$. Furthermore, on the event $\tau \leq r$, the resulting predictor $C(x) = \{y : s(x,y) \leq \tau\}$ has marginal coverage under $Q$ upper bounded by $Q(Y \in C(X)) \leq \theta + (1-\theta)r^d$.*

*Proof.* Let $N := \sum_{i=1}^m \mathbf{1}[X_i \in \mathcal{B}]$ be the number of calibration points falling in $\mathcal{B}$. Because $P(X \in \mathcal{B}) = r^d$ and $m$ is defined as equation 18, it follows that

$$\Pr_{X_{\mathrm{cal}}}[N \geq 1] = 1 - (1 - r^d)^m \geq 1 - e^{-mr^d} \geq 1 - e^{-(c-r^d)}.$$

Now, condition on the event $N \geq 1$. Note that

$$\widehat{F}_m(r) := \frac{N(1 - \theta + \theta/r^d)}{N(1 - \theta + \theta/r^d) + (m - N)(1 - \theta)} \geq \frac{(1 - \theta + \theta/r^d)}{(1 - \theta + \theta/r^d) + m(1 - \theta)},$$

where the last inequality follows since we condition on $N \geq 1$. Next, using $(1 - \theta) + \theta/r^d \geq \theta/r^d$ and $m \leq c/r^d$,

$$\widehat{F}_m(r) \geq \frac{\theta/r^d}{\theta/r^d + m(1 - \theta)} \geq \frac{\theta/r^d}{\theta/r^d + (c/r^d)(1 - \theta)} = \frac{\theta}{\theta + c(1 - \theta)} \geq 1 - \alpha,$$

where the last inequality is due to $c < \frac{\alpha\theta}{(1-\alpha)(1-\theta)}$ in (18). Thus, if $N \geq 1$, then $\tau \leq r$.

Because the score $s(x,y) = \|x\|_\infty$ depends only on $x$, the conformal set is $C(x) = [0,1]$ if $\|x\|_\infty \leq \tau$ and $C(x) = \emptyset$ otherwise. Hence, conditioned on $N \geq 1$, we have

$$Q(Y \in C(X)) = Q(\|X\|_\infty \leq \tau) \leq Q(\|X\|_\infty \leq r) = \theta + (1 - \theta)r^d.$$

$\square$

Letting $r \to 0$ while keeping $\theta$ fixed forces the coverage to converge to $\theta < 1 - \alpha$; the miscoverage is strictly greater than the nominal level $\alpha$. To make this concrete, suppose we choose $\alpha = 0.1$, and $\theta = 0.1$. Then, we can set $c = 0.01$. Proposition 1 then tells us that for $m = 1/r^d$, the output of WCP has a roughly 1% chance of having around 80% miscoverage (independent of $r$ and $d$). In other words, unless the calibration set is on the order of $1/r^d$, WCP cannot guarantee high coverage probability. *Furthermore, we made no attempt to optimize these constants.*

Second, we show the existence of a sample size regime where learned importance weights can catastrophically fail to estimate the importance weights in $L_1$-error. The downstream effect on WCP is a degradation of its *expected* marginal coverage for reasonable sample sizes.

**Proposition 2.** *Fix parameters $d \in \mathbb{N}, r^d \in (0, \theta/4), \theta \in (0, 1/2), 1/\theta \leq m < 1/r^d$. Suppose we draw the source (train) and target (test) sets $X_{\mathrm{train}} = (X_1, \dots, X_m) \sim P^m$ and $X_{\mathrm{test}} = (\tilde{X}_1, \dots, \tilde{X}_m) \sim Q^m$. Then, with probability at least $\frac{1}{e}\left(1 - \frac{1}{e}\right) \geq 0.2325$: $X_{\mathrm{train}} \cap \mathcal{B} = \emptyset$ and $X_{\mathrm{test}} \cap \mathcal{B} \neq \emptyset$. Furthermore, define the class of valid density ratios*

$$w_\beta(x) = \begin{cases} \beta, & x \in \mathcal{B} \\ \frac{1 - r^d\beta}{1 - r^d}, & x \notin \mathcal{B} \end{cases}, \quad \beta \in \left[1 - \theta + \frac{\theta}{r^d}, \frac{1}{r^d}\right]. \tag{19}$$

*If $X_{\mathrm{train}} \cap \mathcal{B} = \emptyset$ and $X_{\mathrm{test}} \cap \mathcal{B} \neq \emptyset$, then $\widehat{R}(w_{\beta'}) < \widehat{R}(w_\beta)$ for all $\beta' > \beta$ (where $\beta, \beta'$ are in the above interval). In other words, if $X_{\mathrm{train}} \cap \mathcal{B} = \emptyset$ and $X_{\mathrm{test}} \cap \mathcal{B} \neq \emptyset$, which occurs with constant probability, then ERM selects the largest possible valid weight for the region $\mathcal{B}$, overestimating the true weight of $1 - \theta + \theta/r^d$. In particular letting $\hat{w}$ denote the learned ratio, the $L_1$ error between $\hat{w}$ and $w^*$ (defined in Example 1) will be $2(1 - \theta)(1 - r^d)$.*

*Proof.* Note that each $X_i$ (resp. $\tilde{X}_i$) lands in $\mathcal{B}$ with probability $r^d$ (resp. $\theta + (1-\theta)r^d$). Thus

$$\Pr_{X_{\text{train}}}[X_{\text{train}} \cap \mathcal{B} = \emptyset] = (1 - r^d)^m$$

$$> (1 - r^d)^{1/r^d} \geq 1/e$$

$$\Pr_{X_{\text{test}}}[X_{\text{test}} \cap \mathcal{B} \neq \emptyset] = 1 - (1 - (\theta + (1-\theta)r^d))^m$$

$$\geq 1 - (1 - \theta)^m \geq 1 - e^{-m\theta} \geq 1 - 1/e.$$

where we have used that $r^d < 1/2$ and $1/\theta \leq m \leq 1/r^d$. Since $X_{\text{train}}$ and $X_{\text{test}}$ are independent,

$$\Pr_{X_{\text{train}}, X_{\text{test}}}[X_{\text{train}} \cap \mathcal{B} = \emptyset \wedge X_{\text{test}} \cap \mathcal{B} \neq \emptyset] = \frac{1}{e}\left(1 - \frac{1}{e}\right).$$

Now, condition on the event $X_{\text{train}} \cap \mathcal{B} = \emptyset \wedge X_{\text{test}} \cap \mathcal{B} \neq \emptyset$.

- Since $X_{\text{train}} \cap \mathcal{B} = \emptyset$, for every training point $X_i$, we have $X_i \notin \mathcal{B}$. Therefore, $w_\beta(X_i) = \frac{1-r^d\beta}{1-r^d}$ for all $i \in [m]$.

- Since $X_{\text{test}} \cap \mathcal{B} \neq \emptyset$, at least one test point $\tilde{X}_j$ falls into $\mathcal{B}$. Let's partition the test set indices into two sets: $I_\mathcal{B} = \{j : \tilde{X}_j \in \mathcal{B}\}$ and $I_{\mathcal{B}^\complement} = \{j : \tilde{X}_j \notin \mathcal{B}\}$. By our conditioning, the set $I_\mathcal{B}$ is non-empty. Let $m_\mathcal{B} = |I_\mathcal{B}| \geq 1$.

We can now write the empirical risk $\widehat{R}(w_\beta)$ as an explicit function of $\beta$:

$$\widehat{R}(w_\beta) = \frac{1}{2}\sum_{i=1}^{m}\left(w_\beta(X_i)^2 - 2w_\beta(\tilde{X}_i)\right)$$

$$= \frac{1}{2}\left[\sum_{i=1}^{m}\left(\frac{1-r^d\beta}{1-r^d}\right)^2 - 2\left(\sum_{j\in I_\mathcal{B}}w_\beta(\tilde{X}_j) + \sum_{j\in I_{\mathcal{B}^\complement}}w_\beta(\tilde{X}_j)\right)\right]$$

$$= \frac{1}{2}\left[m\left(\frac{1-r^d\beta}{1-r^d}\right)^2 - 2\left(m_\mathcal{B}\cdot\beta + (m-m_\mathcal{B})\frac{1-r^d\beta}{1-r^d}\right)\right]$$

To show that $\widehat{R}(w_\beta)$ decreases as $\beta$ increases, we find its derivative with respect to $\beta$:

$$\frac{d}{d\beta}\widehat{R}(w_\beta) = \frac{1}{2}\left[m\cdot 2\left(\frac{1-r^d\beta}{1-r^d}\right)\left(\frac{-r^d}{1-r^d}\right) - 2\left(m_\mathcal{B} + (m-m_\mathcal{B})\frac{-r^d}{1-r^d}\right)\right]$$

$$= -\frac{mr^d(1-r^d\beta)}{(1-r^d)^2} - m_\mathcal{B} + \frac{(m-m_\mathcal{B})r^d}{1-r^d}$$

$$= -\frac{m_\mathcal{B}}{1-r^d} + \frac{m(r^d)^2(\beta-1)}{(1-r^d)^2}$$

We must show this expression is negative. The first term, $-\frac{m_\mathcal{B}}{1-r^d}$, is strictly negative since $m_\mathcal{B} \geq 1$ and $r^d \leq 1$. The second term is positive, since $\beta > 1$. For the derivative to be negative, we need the negative term to have a larger magnitude:

$$\frac{m_\mathcal{B}}{1-r^d} > \frac{m(r^d)^2(\beta-1)}{(1-r^d)^2} \iff m_\mathcal{B}(1-r^d) > m(r^d)^2(\beta-1)$$

Since $m_\mathcal{B} \geq 1$, it is sufficient to show this for $m_\mathcal{B} = 1$:

$$1 - r^d > m(r^d)^2(\beta-1)$$

We use the upper bound for $\beta$: $\beta \leq 1/r^d$. Substituting this in, it suffices to show

$$1 - r^d > mr^{2d}(1/r^d - 1),$$

which is true by assumption that $m < 1/r^d$. Thus, $\frac{d}{d\beta}\widehat{R}(w_\beta) < 0$ for all $\beta \in \left[1 - \theta + \frac{\theta}{r^d}, \frac{1}{r^d}\right]$, which implies the desired claim. $\qquad\square$

Finally, for completeness, we instantiate Corollary 1 on Example 1.

**Proposition 3.** *Let the setting be as in Example 1, with $\mathcal{W}$ defined in Equation* (19). *Consider learning a clipped density ratio $\hat{w}$ and then prediction sets $C_\tau$ as in Corollary 1. Then,*

$$\Pr\left[1 - \alpha \leq \Pr_Q\left[Y \in C_\tau(X)\right] \leq 1 - \alpha + 2\Delta_B + 12\epsilon\right] \geq 1 - 3\delta$$

*where the randomness is over the draw of the density ratio estimation sets, the bias estimation set, and the calibration set. Additionally, we require*

$$\mathcal{O}\left(\frac{B\log(1/\epsilon) + B\log(1/\delta)}{\epsilon^2} + \frac{B^2\log(1/\delta)}{\epsilon^2}\right), \mathcal{O}\left(\frac{B^4 + B^4\log(1/\delta)}{\epsilon^4}\right), \mathcal{O}\left(\frac{B^2 + B^2\log(1/\delta)}{\epsilon^4}\right)$$

*labeled examples from $P$, unlabeled examples from $P$, and unlabeled examples from $Q$, respectively.*

*Proof.* Note that Proposition 3 would follow from Corollary 1 as long as we are able to show that $C_B, \tilde{C}_B = \mathcal{O}(B)$. Let us decompose $\mathcal{W}_B$ as a union of unclipped and clipped components,

$$\mathcal{W}_B = \left\{w_\beta : \beta \in [1 - \theta + \theta/r^d, B]\right\} \cup \left\{\left(x \mapsto \begin{cases} B, & x \in \mathcal{B} \\ \frac{1 - r^d\beta}{1 - r^d}, & x \notin \mathcal{B} \end{cases}\right) : \beta \in [B, 1/r^d]\right\}.$$

Let us refer to the first term as $\mathcal{W}_B^{(1)}$ and the second term $\mathcal{W}_B^{(2)}$. For any $X = (X_1, \ldots, X_m) \in \mathcal{X}^m$,

$$\mathrm{Rad}_X(\mathcal{W}_B) \leq \mathrm{Rad}_X(\mathcal{W}_B^{(1)}) + \mathrm{Rad}_X(\mathcal{W}_B^{(2)}).$$

Thus, we bound each piece independently. To bound the first term, write

$$\mathrm{Rad}_X(\mathcal{W}_B^{(1)}) = \mathbb{E}_{\sigma \sim \{-1,1\}^m}\left[\sup_{\beta \in [1 - \theta + \theta/r^d, B]} \frac{1}{m}\sum_{i=1}^m \sigma_i w_\beta(X)\right].$$

since $w_\beta$ is linear in $\beta$, the maximum will be achieved at an endpoint, where $\beta \in \{1 - \theta + \theta/r^d, B\}$. Thus, $\mathrm{Rad}_X(\mathcal{W}_B^{(1)}) = \mathrm{Rad}_X(\{w_{1-\theta+\theta/r^d}, w_B\}) \leq B/\sqrt{m}$ by Massart's lemma. A similar argument holds for $\mathcal{W}_B^{(2)}$, since $\mathcal{W}_B^{(2)}$ is affinely parameterized by $\beta \in [B, 1/r^d]$, and the maximum must be at the boundary. Massart's lemma again yields $\mathrm{Rad}_X(\mathcal{W}_B^{(2)}) \leq B/\sqrt{m}$. By adding these two bounds, we conclude that $C_B, \tilde{C}_B = \mathcal{O}(B)$ as desired. $\square$

## C  COMPLEXITY BOUNDS FOR CLIPPED CLASSES

Under the assumption that $\mathcal{W}$ has finite combinatorial dimension, we may obtain finer bounds on the Rademacher complexity of $\mathcal{W}_B$. In this section, we present our results for classes with finite fat-shattering dimension, a combinatorial measure which is known to characterize the sample complexity of distribution-independent learning. We define this below.

**Definition 1** (Fat-shattering dimension). *Let $\mathcal{F}$ be a class of real-valued functions on a domain $\mathcal{X}$, and let $\gamma > 0$. We say that a set $S = \{x_1, \ldots, x_m\} \subseteq \mathcal{X}$ is $\gamma$-shattered by $\mathcal{F}$ if there exist real numbers $r_1, \ldots, r_m$ such that for every $\sigma \in \{-1, 1\}^m$ there exists $f \in \mathcal{F}$ satisfying*

$$\sigma_i = 1 \implies f(x_i) \geq r_i + \gamma, \quad \sigma_i = -1 \implies f(x_i) \leq r_i - \gamma, \quad \forall i \in [m].$$

*The $\gamma$-fat-shattering dimension of $\mathcal{F}$, denoted $\mathrm{fat}_\mathcal{F}(\gamma)$, is the largest integer $m$ for which there exists a set of $m$ points that is $\gamma$-shattered by $\mathcal{F}$. If no such largest $m$ exists, then $\mathrm{fat}_\mathcal{F}(\gamma) = \infty$.*

**Example 2.** *Let $\mathcal{F}$ be the class of linear functions over $\mathbb{R}^d$. Then, $\mathrm{fat}_\mathcal{F}(\gamma) = d$.*

We rely on the property that clipping does not increase the fat-shattering dimension of $\mathcal{F}$. We prove this below for completeness.

**Lemma 10.** *Let $\mathcal{F} \subseteq \mathbb{R}^\mathcal{X}$. Define the clipped class $\mathcal{F}_B = \{x \mapsto \max(\min(f(x), B), -B) : f \in \mathcal{F}\}$. Then for any $0 \leq \gamma \leq B$, it holds that $\mathrm{fat}_{\mathcal{F}_B}(\gamma) \leq \mathrm{fat}_\mathcal{F}(\gamma)$.*

*Proof.* Suppose $S = \{x_1, \ldots, x_m\}$ is $\gamma$-shattered by $\mathcal{F}_B$, and let $r_1, \ldots, r_m$ be the witness. For every $\sigma \in \{-1, 1\}^m$, let $f_B^\sigma \in \mathcal{F}_B$ be a function satisfying

$$\sigma_i = 1 \implies f_B^\sigma(x_i) \geq r_i + \gamma, \quad \sigma_i = -1 \implies f_B^\sigma(x_i) \leq r_i - \gamma, \quad \forall i \in [m].$$

Clearly, it must be that $-B + \gamma \leq r_i \leq B - \gamma$, or else the above implications could not be satisfied, since the range of functions in $\mathcal{F}_B$ is $[-B, B]$. Now, let $f \in \mathcal{F}$ and define $f_B(x) = \max(\min(f(x), B), -B)$. It can be easily checked that

$$f_B(x) \geq r + \gamma \implies f(x) \geq r + \gamma, \quad f_B(x) \leq r - \gamma \implies f(x) \leq r - \gamma, \quad \forall r \in [-B + \gamma, B - \gamma].$$

On the other hand, since each $f_B \in \mathcal{F}_B$ can be written like this, it follows that any sign behavior that can be expressed by $\mathcal{F}_B$ with witnesses in the range $[-B + \gamma, B - \gamma]$ can also be expressed by $\mathcal{F}$. In particular, we use apply this to the functions $f_B^\sigma$ and conclude that $\mathrm{fat}_{\mathcal{F}_B}(\gamma) \leq \mathrm{fat}_{\mathcal{F}}(\gamma)$. $\square$

Equipped with this lemma, we can now derive an explicit bound on the Rademacher complexity of $\mathcal{F}_B$ in terms of $B$ and the fat-shattering dimension of $\mathcal{F}$. For ease of exposition, we assume that the fat-shattering dimension is upper bounded by a constant as $\gamma \to 0$ (which is the case for Example 2 and more generally, classes with finite pseudodimension).

**Proposition 4.** *Let $\mathcal{F} \subseteq \mathbb{R}^{\mathcal{X}}$ define $\mathcal{F}_B$ as in Lemma 10. Assume that $\mathrm{fat}_{\mathcal{F}}(\gamma) \leq d$ for all $\gamma > 0$. Then for every sample $X = (X_1, \ldots, X_m) \in \mathcal{X}^m$ the empirical Rademacher complexity satisfies*

$$\mathrm{Rad}_X(\mathcal{F}_B) = \mathcal{O}\left(B\sqrt{\frac{d}{m}}\right).$$

*Proof.* We begin with an application of chaining; by Theorem 1.1 of Kakade & Tewari (2008), for any sample $X = (X_1, \ldots, X_m) \subseteq \mathcal{X}^m$, we may bound the empirical Rademacher complexity by

$$\mathrm{Rad}_X(\mathcal{F}_B) \leq 12 \int_0^\infty \sqrt{\frac{\log N_2(\alpha, \mathcal{F}_B, X)}{m}} \, d\alpha$$

$$= \frac{12}{\sqrt{m}} \int_0^B \sqrt{\log N_2(\alpha, \mathcal{F}_B, X)} \, d\alpha, \qquad (F_B \text{ has range in } [-B, B])$$

where $N_2(\alpha, \mathcal{F}, X)$ is the $L_2$-covering number of $\mathcal{F}_B$ on the sample $X$. On the other hand, from Theorem 1 of Mendelson & Vershynin (2003) (after suitable rescaling by $1/B$) along with Lemma 10 we may bound the log covering number as

$$\log N_2(\alpha, \mathcal{F}_B, X) \leq C_1 \mathrm{fat}_{\mathcal{F}}(C_2\alpha) \log(B/\alpha), \quad \forall \alpha \in [0, B]$$

for some universal constant $C_1, C_2 > 0$. Combining with the above integral, we conclude

$$\mathrm{Rad}_X(\mathcal{F}_B) = \mathcal{O}\left(B\sqrt{\frac{d}{m}}\right).$$

$\square$

**Remark 6.** *In particular, we may instantiate this with linear classes to derive a regime where clipping yields a significant reduction in the Rademacher complexity of $\mathcal{F}$. Let $\mathcal{F} = \{x \mapsto y^\top x : y \in \mathbb{R}^d, \|y\|_2 \leq U\}$. By Proposition 4 and Example 2, we have that $\mathrm{Rad}_X(\mathcal{F}_B) = \mathcal{O}(B\sqrt{\frac{d}{m}})$. On the other hand, by directly bounding the Rademacher complexity, and then applying Talagrand's contraction principle, we may obtain $\mathrm{Rad}_X(\mathcal{F}_B) \leq \mathrm{Rad}_X(\mathcal{F}) \leq UR/\sqrt{m}$ assuming $\|X_i\|_2 \leq R$ for all $i \in [m]$. Thus, Proposition 4 reveals a regime where $B \leq UR/\sqrt{d}$ where clipping allows a significantly sharper bound on the complexity of $\mathcal{F}_B$ than the naive strategy in Remark 1.*

# D  CLISF WITH PIECEWISE CONSTANT DENSITY RATIOS

In this section, we assume that the input space $\mathcal{X}$ consists of points of the form $(X^0, X^1, Y)$ where $X^0 \in [k]$ is a subpopulation identifier, $X^1$ contains additional covariate information, and $Y$ is the outcome. We assume that $P$ has the form

$$X^0 \sim \mathrm{Multinomial}(p_1, \ldots, p_k), \quad (X_1, Y) \mid (X^0 = i) \sim \Pi_i$$

i.e., the training data point is drawn from group $i$ with probability $p_i$, and then conditional on being drawn from group $i$, the remaining features $X^1$ and outcome $Y$ are drawn from some joint distribution $\Pi_k$. We assume that $Q$ has the form

$$X^0 \sim \text{Multinomial}(q_1, \ldots, q_k), \quad (X_1, Y) \mid (X_0 = i) \sim \Pi_i.$$

In other words, $P$ and $Q$ are both mixtures of the $\Pi_i$, but with different mixture weights. Thus, the true weights have a piecewise constant structure, where $w^*(X^0, X^1, Y)$ depends only on the subpopulation identifier $X^0$. This is the setting considered by Bhattacharyya & Barber (2024). This also subsumes the setting of Appendix B of Park et al. (2021), by taking $X_0 = j(X^1)$, where $j : \mathcal{X} \to [k]$ is some clustering model. Park et al. (2021) propose to use bucketed source discriminators or unsupervised learning to estimate the clusters.

In this setting, we consider two very natural settings of the density ratio class $\mathcal{W}$ and show that each leads to efficient optimization of the CLISF objective.

**Unknown train distribution.** We consider the class $\mathcal{W}$ of piecewise constant weights $w(X^0, X^1) = w_i \in \mathbb{R}_+$ for $X^0 = i$. In this case, the empirical CLISF objective, over a sample $X_{\text{train}} = (X_1, \ldots, X_m)$ and $X_{\text{train}} = (\tilde{X}_1, \ldots, \tilde{X}_m)$, is equivalent to the convex QP

$$\text{Minimize} \quad \frac{1}{2} \sum_{i=1}^m \left( w_{X_m^0}^2 - 2 w_{\tilde{X}_m^0} \right) \quad \text{over } w_1, \ldots, w_k \in \mathbb{R}$$

$$\text{Subject to} \quad 0 \leq w_i \leq B, \quad \forall i \in [k]$$

and hence may be solved efficiently. Since there are no second-order interactions between the different $w_i$, this may be minimized pointwise for each $w_i$ by taking $w_i = \min(\tilde{m}_i / m_i, B)$ where $m_i$ is the number of training points falling in cluster $i$, and $\tilde{m}_i$ is defined similarly for the test points. When $m_i = 0$, we follow the convention that $\tilde{m}_i / m_i = \infty$.

**Known train distribution.** Now, assume the train marginal $P_X$ is known. More specifically, assume we have access to the mixture weights $p_1, \ldots, p_k$. We can incorporate this information into an additional affine constraint on our feasible set, which enforces that the density ratios cannot integrate to more than 1 under $P$:

$$\text{Minimize} \quad \frac{1}{2} \sum_{i=1}^m \left( w_{X_m^0}^2 - 2 w_{\tilde{X}_m^0} \right) \quad \text{over } w_1, \ldots, w_k, b_1, \ldots, b_k \in \mathbb{R}$$

$$\text{Subject to} \quad 0 \leq w_i \leq B, \quad b_i \geq 0, \quad \forall i \in [k]; \quad \sum_{i=1}^k p_i(w_i + b_i) = 1$$

where $b_1, \ldots, b_k$ are slack variables representing the clipping bias. This is another convex QP in $w_1, \ldots, w_k, b_1, \ldots, b_k$ and hence may be solved efficiently.

# E    OTHER APPROACHES TO CHOOSING THE CLIPPING PARAMETER

In this section, we discuss additional strategies to select the clipping parameter $B$.

**Choosing $B$ via Corollary 1.** Consider fixing the sample sizes. The dominant dependence on $B$ for the sample sizes in Corollary 1 are for the unlabeled $P$ and $Q$ examples. Assuming that $C_B, \tilde{C}_B = \mathcal{O}(B^p)$, we may invert these sample sizes to obtain a heuristic $B \approx m^{\frac{1}{2(p+1)}} \epsilon^{\frac{2}{p+1}}$. However, this may be overly conservative and in practice it suffices to choose a larger value of $B$.

**Choosing $B$ to make $\Delta_B$ small.** A natural question is whether we can precisely control $\Delta_B$ in terms of $B$. If we choose $B$ large enough such that $\Delta_B = \mathcal{O}(\epsilon)$, then in (4), the overcoverage will not depend on $\Delta_B$; this is an "unbiased" coverage guarantee. Furthermore, if $\inf\{B : \Delta_B \leq \epsilon\} = \text{poly}(1/\epsilon)$, then the sample size is polynomial in $1/\epsilon$. However, precisely controlling $\Delta_B$ is not possible in general. For example, consider a two-symbol universe $\{a, b\}$, where $P(\{a\}) = p$ and $P(\{b\}) = 1 - p$, and $Q$ is uniform over $\{a, b\}$. If $p \to 0$, then $\inf\{B : \Delta_B \leq \epsilon\} \to \infty$ for any fixed $\epsilon$. To obtain rate control in $B$, we thus assume additional tail penalization on $w^*$. The below proposition applies, for example, with the $\chi^2$ distance when $P$ and $Q$ are known to be spherical Gaussians with similar variance (Corollary 1 of Rubenstein et al. (2019)).

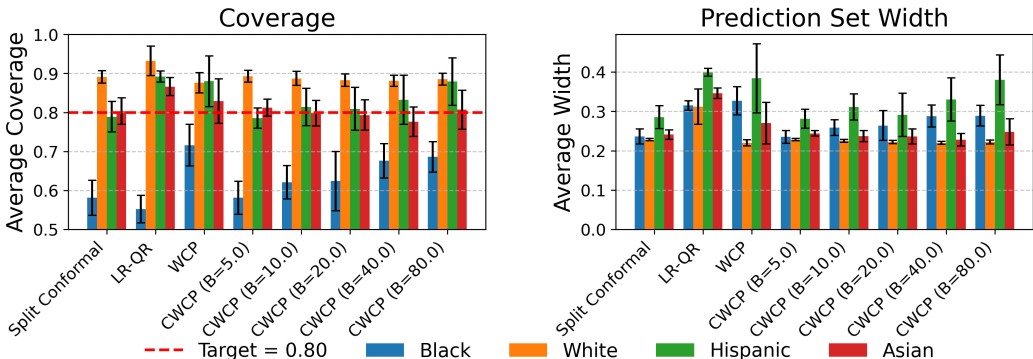

Figure 4: Coverage results for CWCP ($B \in \{5, 10, 20, 40, 80\}$), split conformal, WCP, and LR-QR on Communities and Crime data. The colored bars represent average coverage and prediction set size for each algorithm and the black bars represent $\pm 1$ standard deviation.

**Proposition 5.** *Let $f : \mathbb{R}_+ \to \mathbb{R}_+$ be nondecreasing on $[B_0, \infty)$. Let $\rho := \mathbb{E}_P[f(w^*(X))]$. Then, $\Delta_B := \mathbb{E}_P[(w^*(X) - B)^+] \leq \rho \cdot \int_B^\infty \frac{1}{f(t)} \, dt$ for all $B \geq B_0$. In particular, if $f(x) \geq C(x - B_0)^p$ for all $x \geq B_0$, for some $C > 0$ and $p > 1$, then $\Delta_B \leq \frac{\mathbb{E}_P[f(w^*(X))]}{C(p-1)(B-B_0)^{p-1}}$ for all $B \geq B_0$.*

*Proof.* By Markov's inequality, and using the assumption that $f$ is nondecreasing, for any $\alpha \geq B_0$,

$$\Pr_P [w^*(X) \geq \alpha] \leq \Pr_P [f(w^*(X)) \geq f(\alpha)] \leq \frac{\mathbb{E}_P[f(w^*(X))]}{f(\alpha)} = \frac{\rho}{f(\alpha)}.$$

By integrating this upper bound on the tail probability, we find

$$\Delta_B := \mathbb{E}_P[(w^*(X) - B)^+] = \int_B^\infty \Pr_P [w^*(X) \geq t] \, dt \leq \rho \cdot \int_B^\infty \frac{1}{f(t)} \, dt.$$

To prove the second part of the claim, we use the assumption that $f(x) \geq C(x - B_0)^p$, which implies $1/f(x) \leq \frac{1}{C(x-B_0)^p}$. This argument yields

$$\Delta_B \leq \frac{\mathbb{E}_P[f(w^*(X))]}{C} \cdot \int_B^\infty \frac{1}{(t - B_0)^p} \, dt = \frac{\mathbb{E}_P[f(w^*(X))]}{C(p-1)(B - B_0)^{p-1}}, \quad \forall B \geq B_0.$$

$\square$

# F  ADDITIONAL EXPERIMENTS

## F.1  COMMUNITIES AND CRIME

We additionally evaluate our methods on the *Communities and Crime* dataset Redmond (2002), which contains 1994 datapoints of communities in the United States, each datapoint being a 127-dimensional input. The task is to predict the violent crime rate. Following Joshi et al. (2025), We first randomly select half of the data as a training set, and use it to fit a 1 hidden layer neural network as our predictor. We use the remaining half to design four covariate shift scenarios, determined by the frequency of a specific racial subgroup. For each of these features, we find the median value $m$ over the remaining dataset. Datapoints with feature value at most $m$ form our source set, and the rest form our target set. This creates a covariate shift between train and test datasets.

**Experimental details.** The nonconformity score is the residual to our regression model. The we considered $\mathcal{W}$ defined by linear maps from the features space to $\mathbb{R}$. We ran 30 trials in total, with a coverage target of $1 - \alpha = 0.8$, as in Joshi et al. (2025). For each trial, we measured the coverage on a held out test set as well as the width of the resulting prediction interval. In contrast to Joshi et al. (2025), who considered a ratio class consisting of linear maps directly from the feature space to $\mathbb{R}$, we considered the class of linear maps from the hidden layer of the regression model.

**Results.** Figure 4 displays the results. For the Hispanic and Asian population covariate shifts, CWCP achieved both average coverage close to $0.8$ as well as low coverage variance. LR-QR also achieved stable coverage. On the other hand, WCP had very high variance on the Hispanic and Asian shifts. As predicted by our theory, the amount of variation tended to increase with $B$. For the White population covariate shift, all methods slightly overcovered. Interestingly, WCP achieved a slightly lower overcoverage compared to other methods, although with a higher variance in coverage.

Next, for the Black population shift, all methods except for WCP and CWCP (with high $B$) seemed to greatly undercover. For the density ratio-based methods (WCP, LR-QR, and CWCP) a possible explanation is that the class of ratios did not correctly capture the nature of the covariate shift in this case, leading to high misspecification. For split conformal, a likely explanation is that it did not take the covariate shift into account.

Regarding set sizes, for the Black, Hispanic, and Asian population covariate shifts, split conformal and CWCP ($B = 5$) appeared to produce the smallest prediction sets on average. This is not surprising, as split conformal and CWCP ($B = 5$) tended to exhibit less overcoverage compared to other methods, particularly on the Hispanic and Asian shifts. In contrast, LR-QR, WCP, and CWCP ($B = 80$) had the most overcoverage and, unsurprisingly, also the largest prediction set widths. A key takeaway is that the good coverage performance of CWCP *does not* rely on outputting trivial prediction sets, as evidenced by the relatively low prediction set widths.

## F.2 EMPIRICAL VALIDATION OF SRM FOR CLIPPING PARAMETER SELECTION ON SYNTHETIC DATA

We additionally investigate the performance of a SRM-based strategy for selecting $B$. As a proof of concept, we implement a structural risk-regularized objective on the synthetic data setting from Section 5.2. For varying sample sizes, we will investigate the generalization behavior of the empirical minimizer of a SRM-regularized CLISF objective.

**Experimental details.** We consider the same distributions and density ratio class as Section 5.2. In fact, since we are only interested in the density ratio estimation part (CLISF) of the CWCP pipeline, we need only consider the marginal covariate distributions of $P$ and $Q$. Thus, the task is equivalent to estimating the density ratio between two shifted Gaussians. We used $d = 200$ in our experiments and considered a fixed shift magnitude of $\beta = 2$ (this choice was arbitrary).

The SRM-regularized CLISF objective we solved was

$$\arg \min_{B \in \{2.5, 5, 10, 20, 40\}, w \in \mathcal{W}_B} \widehat{R}(w) + \lambda \cdot B \sqrt{\frac{d}{m}},$$

where $\widehat{R}(w)$ is as in (3) and $\lambda \cdot B \sqrt{\frac{d}{m}}$ denotes the complexity regularization term chosen per Appendix C, with $\lambda \geq 0$ denoting a regularization strength. We ran our experiments with varying choices $\lambda \in \{0, 0.1, 0.3, 0.5, 0.7, 0.9, 1\}$ and varying sample sizes $m \in \{50, 100, \ldots, 500\}$. We ran 100 trials and measured the average generalization performance (in terms of the population square loss $\mathbb{E}_P[(\hat{w}(X) - w^*(X))^2]$) for each combination of $B$, $\lambda$, and $m$.

**Results.** Figure 5 displays the results. The bottommost figure plots the average test performance (in terms of the population square loss $\mathbb{E}_P[(\hat{w}(X) - w^*(X))^2]$) of the learned clipped density ratio against the sample size $m$. Different colors indicate different choices of $B$, and the shaded colored regions indicate $\pm 1$ standard deviation. The top six plots (each representing a value of $\lambda$) represent the value of the SRM-regularized CLISF objective, again against the sample size $m$. Qualitatively, the best choice of regularizer $\lambda$ will correspond to a plot which most closely matches the bottommost plot (corresponding to the test losses): this indicates that the best choice of $B$ according to the SRM-regularized objective is close to the best choice of $B$ if we had known the test losses in advance. This is clearly achieved for $\lambda = 0.5$, which very closely tracks the test loss plot.

For lower values of $\lambda < 0.5$, we observe that there was insufficient penalty for structural complexity. This is because the lowest training loss was attained by the highest value of $B = 80$, whereas this value achieved the worst generalization performance until $m \approx 200$, and did not become competitive with the best choice of $B$ until $m \approx 500$. This is a clear sign of overfitting due to $\lambda$ being insufficiently large.

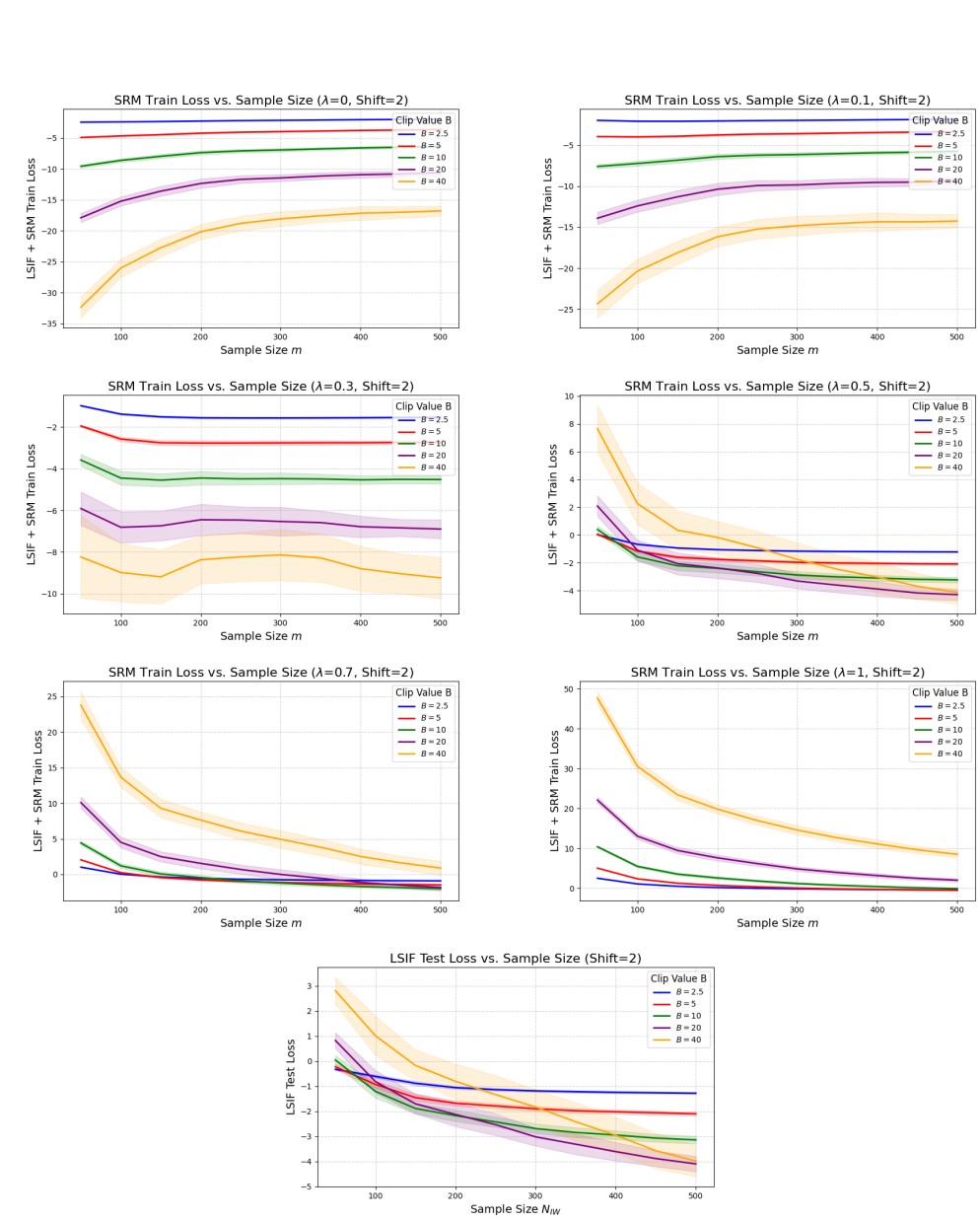

Figure 5: Results for the structural risk-regularized CLISF objective. Qualitatively, the best choice of regularizer $\lambda$ will correspond to a plot which most closely matches the bottommost plot: this is clearly attained when $\lambda = 0.5$.

For higher values of $\lambda > 0.5$, we observe that there was too much penalty for structural complexity. This is evidenced by the fact that the SRM-regularized objective favored smaller values even for higher sample sizes. For example, when $\lambda = 0.7$, the green curve (corresponding to $B = 10$) did not go below the red and blue curves ($B = 2.5, 5$) until $m \approx 400$, much later than on the test loss plot. On the other hand, at least in this example, the suboptimality due to an overly conservative choice of $\lambda$ appears relatively benign, especially for lower values of $m$ where only the yellow curve ($B = 40$) was significantly higher than the others.

However, this approach has limitations. First, it exchanges the problem of selecting $B$ for the problem of selecting the regularization strength $\lambda$. While $\lambda$ is a universal constant related to the Rademacher complexity constants, in practice, the theoretical bounds are often loose, requiring $\lambda$ to be tuned as a hyperparameter. Nevertheless, our experiments suggest that a single choice of $\lambda$ (e.g., $\approx 0.5$) is robust across varying sample sizes, unlike $B$, which must strictly grow with $m$. Second, the computational cost is higher than a single fit, as one must solve the CLISF objective for a grid of $B$ values to identify the minimum of the penalized risk profile.

Our empirical results suggest that SRM provides a robust, data-driven mechanism for navigating the bias-variance tradeoff. Crucially, while the optimal clipping threshold $B$ shifts dramatically with sample size (as seen in the bottom panel), the optimal regularization strength $\lambda \approx 0.5$ remains stable across the entire range of $m$. This implies that SRM effectively transforms the difficult problem of selecting a dynamic, sample-dependent parameter $B$ into the simpler task of selecting a static, structural constant $\lambda$. By penalizing the hypothesis complexity directly, the method allows the estimator to automatically adapt its capacity to the available data, tracking the optimal test performance without requiring access to the target labels.

## G LLM USAGE STATEMENT

The authors used AI tools (GPT-5 and Gemini 2.5 Pro) as an aid in writing code, surveying related literature, and providing feedback on the manuscript, with careful instructions from the authors. All LLM-produced content was reviewed and edited by the authors before usage.

