# OpenReview forum: "Weight Clipping for Robust Conformal Inference under Unbounded Covariate Shifts"
_ICLR.cc/2026/Conference — Submitted to ICLR 2026_

### Official Review · Reviewer_wftD · 2025-10-15

**Soundness:** 2
**Presentation:** 2
**Contribution:** 2
**Rating:** 4
**Confidence:** 4

**Summary:**

This paper addresses a critical failure mode of WCP, which is used to provide valid prediction intervals under covariate shift.  WCP becomes unstable and can suffer from significant undercoverage  (but also overcoverage) when the true density ratio between the source and target distributions is unbounded or must be learned from data, leading to high-variance estimates. The authors propose Clipped Weighted Conformal Prediction (CWCP). The method consists of two main parts: 1) Clipped Least-Squares Importance Fitting (CLISF): Instead of learning an unbounded density ratio, they learn a ratio that is explicitly clipped at a threshold $B$, which introduces biases but reduces the variance of the estimator. 2) The authors provide a method to estimate the bias $\Delta_B$ introduced by this clipping directly from the data. They then correct for this bias by running WCP with a slightly inflated coverage target $1 - \alpha - \Delta_B$. The key contribution is providing theoretical guarantees for this weight-clipping approach. They prove that CWCP achieves dataset-conditional coverage with a sample complexity that, crucially, does not depend on the higher moments of the true density ratio.

**Strengths:**

1. **Addresses a Significant and Practical Problem:** The key contribution is providing theoretical guarantees for this weight-clipping approach. They prove that CWCP achieves dataset-conditional coverage with a sample complexity that, crucially, does not depend on the higher moments of the true density ratio.
2. **Elegant Core Idea:** The idea of clipping weights is intuitive, and while it has been proposed already (implicitly or explicitly) in many WCP-related works, the paper's strength lies in formalizing it into a complete framework. The ability to estimate the clipping-induced bias and use it to correct the coverage level is a strong and elegant contribution, even if the underlying theory has limitations.
3. **Novel Theoretical Analysis:** The core theoretical achievement is deriving PAC-style guarantees for a clipped importance weighting scheme that circumvents the dependence on the higher moments of the true density ratio. This directly addresses the primary failure mode of standard WCP.

**Weaknesses:**

1. **Reliance on a Cascade of Unrealistic and Problematic Assumptions:** The paper's theoretical guarantees are built on a foundation of assumptions that are individually and collectively problematic for practical application.
    1. **Unrealistic "No Misspecification" Assumption (Assumption 4):** The main theorems assume the true density ratio w* is perfectly contained within the chosen model class W $(\Delta_R = 0)$. This "realizability" assumption is almost never true in practice, yet this primary source of error is assumed away for "simplicity," making the final guarantees appear stronger than they would be in reality.
    2. **Circular Logic and Arbitrary Constants in Bias Assumptions:** The theory requires the clipping bias $\Delta_B \leq 1/2$ (Assumption 3) and the learning error $\epsilon \leq 1/4$ (Theorem 3). These are presented as fixed preconditions, but both $\Delta_B$ and $\epsilon$ are unknown outcomes of the modeling process. This creates a circular dependency and relies on arbitrary constants, turning critical preconditions into unverifiable a posteriori checks.
    3. **Comparison with dTV:** I’m asking myself, after all this complex theoretical machinery, have we actually gained anything practical over just using split conformal and accepting some degradation, or using a simpler correction based on a single assumption (which you don’t evaluate in your experiments)?
2. **Insufficient and Potentially Misleading Experimental Evaluation:** The experiments fail to provide a complete picture of the method's practical utility.
    1. **Unfair Baseline Comparisons:** The paper compares CWCP, which has stronger dataset-conditional (PAC) guarantees, against methods like standard WCP that only provide weaker marginal guarantees. This is an "apples-to-pears" comparison that makes CWCP appear more stable by design, without comparing it to other methods that offer similar PAC-style guarantees.
    2. **Missing Critical Metrics:** The evaluation focuses exclusively on coverage, ignoring the "efficiency" of the prediction sets (i.e., their average size or width). Without this, it is impossible to know if the method achieves coverage by producing usefully tight intervals or trivially large ones. This is particularly concerning given the risk of over-correction.
    3. **Incomplete Ablation Study:** The ablation study only varies the clipping parameter B, ignoring the many other critical (and difficult to set) modeling choices, such as the choice of the function class $W$.
    4. **Suboptimal Visualization:** The empirical CDF plots used to show coverage distribution are difficult to interpret. Standard visualizations like box plots or violin plots would communicate the variance and central tendency of the coverage much more clearly.
3. **Example 1:** I find it a bit weird that you say the problem is not that theoretical; however, you proceed with a highly artificial example to show that the generalization bound fails when the error has higher moments.
4. **Practical Barriers to Implementation:** The combination of several challenges limits the method's off-the-shelf usability
    1. the non-convex nature of the CLISF optimization problem;
    2. the lack of a concrete, data-driven procedure for selecting the critical hyperparameter;
    3. the assumption of a continuous score CDF, which is presented as standard but often fails to hold in practice;
    4.  the risk that the bias correction term becomes so large that it results in trivially large, uninformative prediction sets.

**Questions:**

- **Baselines & Fairness:** Why did you not compare against other methods that also provide dataset-conditional guarantees? Furthermore, what value of δ was used for your method's implementation in the experiments?
- **Efficiency Metrics & Trivial Sets:** Could you provide results on the average prediction set size for all methods in your experiments? This is crucial for understanding whether CWCP provides meaningful, efficient prediction sets. Did you observe any instances in your experiments where the correction term $\Delta_B + 5\epsilon$ was large enough to cause the target quantile to exceed 1, leading to these uninformative sets?
- **Ablation Studies:** Can you comment on the sensitivity of your method to the choice of the function class $W$? An ablation study showing the impact of model misspecification would significantly strengthen the paper's practical claims.
- **Theoretical Assumptions:** Your main theorems assume no model misspecification $(\Delta_R =0)$. Could you provide the full statement of Theorem 3 where the guarantee explicitly depends on $\Delta_R $? The guarantee in Theorem 3 depends on several assumptions with fixed constants ($\Delta_B \leq 1/2, \epsilon \leq 1/4$), but these quantities are unknown in practice. Could you comment on the choice to use fixed constants rather than stating the bounds in terms of these error parameters directly?
- **PAC-boundry:** maybe I missed it but could you state explicitly that you use PAC-boundry to enable training conditional coverage guarantes and that this is already worked out and formalized by Park et al. (2021) and Pournaderi and Xiang (2024) under coveriate shift.
- Some minor remarks:
    - line 046: “*First, unbounded ratios lead to high-variance estimates of the coverage threshold and reduce the “effective sample size” (Tibshirani et al., 2019).”*
        - note that every deviation form the uniform weights will reduce the effective sample size
    - line 964;: state maybe ERM in full first
    - line 297: “*Our analysis relies on a standard assumption in conformal prediction, that the CDF of the nonconformity scores to be continuous.”* Is it really? I don’t think so.

---

> ### Author Response · Authors · 2025-11-21
>
> (1/2)
>
> We appreciate the detailed and constructive review. We have addressed your concerns regarding assumptions and metrics in the revised manuscript.
>
> **1. Assumption of ``No Misspecification'' (Assumption 4):**
> You correctly pointed out that assuming $w^* \in \mathcal{W}$ is unrealistic. We have generalized our theoretical results. **Theorem 2** and **Corollary 1** in the revised paper now explicitly account for model misspecification error, denoted as $\Delta_R$. The bounds now include an undercoverage term scaling with $\sqrt{\Delta_R}$. This undercoverage term vanishes under the original assumption that $w^* \in \mathcal{W}$. We have also included a discussion of the dependence of our guarantees on the misspecification error in **Section 4.2**.
>
> **2. Circular Logic / Constants:** To clarify, in our main results $\epsilon$ should be thought of as a user-chosen error parameter which can be driven to zero (in the case of no misspecification) or $O(\sqrt{\Delta_R})$ (in the case of misspecification) by increasing the sample size used in **Algorithm 1** (see **Theorem 1**). In **Theorem 2 and Theorem 3**, we treat $\epsilon$ as fixed as we assume that we have already obtained the density ratio estimate $\hat w$ somehow; this allows us to provide a more general statement of **Theorem 2 and Theorem 3** which does not rely on *how* we obtained the estimate $\hat w$. On the other hand, in **Corollary 1** we provide end-to-end guarantees of the CLISF + CWCP pipeline; here, we again treat $\epsilon$ as a user-specified parameter that can be taken to zero by increasing the sample budget. Thus, there is no circular dependency.
>
> On the other hand, you are correct about $\Delta_B$ being an uncontrollable outcome of the modeling process, *for a fixed value of $B$*. However, a key point of our theory is that we are able to estimate $\Delta_B$ and correct the bias so our prediction sets do not undercover (**Theorem 2 and Theorem 3**). Additionally, in **Section 3.2 and Appendix E**, we discuss approaches to choosing $B$; in particular, our **Proposition 5** in the appendix gives general conditions for the bias $\Delta_B$ to decay rapidly (polynomially) as a function of $B$.
>
> **3. Comparison with split conformal / dTV:** Split conformal can suffer from undercoverage as high as $TV(P,Q)$. To fix this, one needs to estimate TV distance, which is notoriously difficult (requires solving a hard variational problem or density estimation). To illustrate our point, we have included a new **Section 4.3** discussing the tradeoffs between split conformal, WCP, and our method. We argue that for a broad class of heavy-tailed covariate shifts, the following hold simultaneously: (i) the total variation distance $\text{TV}(P, Q)$ is high, ruling out split conformal due to high undercoverage, (ii) the second moment $\mathbb E_P[w^*(X)^2]$ is infinite, ruling out WCP due to high variance, and (iii) with a very moderate choice of clipping parameter, the clipping bias can be driven to zero *and* the variance can be controlled. These lend credence to the practical applicability of CWCP under realistic, heavy-tailed covariate shifts.
>
> **4. Baselines and Fairness:** We compared against **Likelihood-Regularized Quantile Regression (LR-QR)** (Joshi et al., 2025), which is a very recent state-of-the-art method addressing similar issues. In particular, they also provide dataset-conditional guarantees (see their Theorem 4.3). Additionally, for nonconformity score-based CP methods like WCP or split conformal, marginal coverage guarantees can be converted into dataset-conditional guarantees via concentration inequalities. However, we emphasize that this is **difficult** for WCP because its learned ratios have a *high range* or *high variance*. We chose to highlight that standard WCP *is* the primary competitor because CWCP is proposed as a direct, robust replacement for it.

---

> ### Author Response · Authors · 2025-11-21
>
> (2/2)
>
> **5. Regarding set sizes and efficiency metrics:** In **Appendix F.1 (Figure 4)**, we analyze the Communities and Crime dataset and report both coverage and **prediction set width**, although we did not provide commentary on the widths in the initial submission. We have added explicit commentary on the observed set sizes in **Appendix F.1**. We observed a trade-off: for a fixed sample size, very small $B$ (high bias) leads to tighter sets but undercoverage, while very large $B$ recovers WCP behavior (good coverage on average, but high variance and potentially very large sets due to unstable weights). Additionally, we did rarely observe trivial sets in practice, but as the standard deviation bars on **Figure 4** indicate, the strong majority of prediction sets produced by our algorithms were nontrivial.
>
> Regarding the possibility of trivial sets, in contrast to other solutions our method provides a *two-sided* coverage guarantee: that is, under the settings of **Theorem 2 and Theorem 3**, our overcoverage is bounded by $\Delta_B$, which in turn is bounded by $\text{TV}(P, Q)$. Note that, if $\Delta_B$ is large then so too is $\text{TV}(P, Q)$. In general no method can hope to achieve both zero undercoverage and overcoverage $\leq \text{TV}(P, Q)$, as in the worst case the difference could be due to (effectively) disjoint supports. On the other hand, for many classes of covariate shift the bias can in fact be reduced at a reasonable rate by increasing $B$; see the **Circular logic / constants** section of this comment for additional details.
>
> **6. Regarding an ablation study on the choice of density ratio class:** We acknowledge that an ablation study could significantly strengthen our empirical claims. Due to time constraints we have not yet run these experiments. However, our findings do present some evidence of the framework's insensitivity to choice of model class. For example, in our iWildCam experiment, we chose the class of ratios simply based on linear heads over the representation layer of a widely used pretrained model (ResNet-18), with no other modifications.
>
> **7. Nonconvexity and computational hardness:** We acknowledge that the nonconvexity of the problem presents a thorn for the theoretical guarantees. However, we note that in many cases, the original LSIF problem was already nonconvex. Particularly with neural networks, CLISF can be implemented by simply adding a clipping layer as a final activation. Additionally, our experiments show that the nonconvexity is not a major issue in practice, as CWCP was able to achieve competitive guarantees on iWildCam, Communities and Crime, and synthetic data.
>
> **8. Regarding PAC-boundary:** We acknowledge that Park et al. (2021) and Pournaderi and Xiang (2024) provide similar PAC/dataset-conditional guarantees under covariate shift. However, a key advantage of our framework is that we prove a *two-sided* guarantee, i.e., in addition to achieving the desired coverage we *do not significantly overcover*. On the other hand, Park et al. (2021) only *empirically* validate that they have low overcoverage and set sizes.
>
> **9. Continuity Assumption:** We acknowledge the comment on the continuity of the score CDF. We have clarified that the continuity assumption is **only needed for establishing upper bounds on coverage** and is not needed for the lower bound. The continuous score CDF is standard in CP theory for establishing upper bounds as it is needed to ensure that quantiles of the nonconformity score are uniquely defined (Lei et al. 2018).
>
> **10. Clarifications:**
> - **$\delta$ value:** We did not explicitly use a $\delta$ value or $\epsilon$ in our experiments. Instead, since we were working with fixed datasets (iWildcam and Communities and Crime), we treated the sample size as fixed.
> - **Effective Sample Size:** We made a small edit to emphasize that unbounded ratios led to *large* decreases in the effective sample size.

---

### Official Review · Reviewer_Nxh1 · 2025-10-15

**Soundness:** 2
**Presentation:** 3
**Contribution:** 1
**Rating:** 2
**Confidence:** 3

**Summary:**

To address extreme density ratio weights in weighted conformal prediction (WCP), the authors proposed a weight clipping method by bounding the weight by a pre-specified constant $B$.

**Strengths:**

The paper is well-written. Formulas and notation are well-typed and shaped nicely. Experiments are designed well and the presentations are very clear.

**Weaknesses:**

I acknowledge the theoretical investigations, efforts and contributions by the authors. This work is publishable given its **practical novelty.** However, at least at this time point, I don't think ICLR is a reasonable choice for this work, since I believe the novelty is not at a fundamental level that can bring much interest for future research. The contribution is instead minor for addressing an existing but less important issue.

Overall, the impression of the paper to me is like modifying the implementation details of the density ratio weighting, instead of focusing on a novel setting or new problem. The techniques used in this paper are also not too novel. The clipping or stabilizing ideas are pretty common, such as propensity score trimming and truncation when addressing extreme inverse probability weights in treatment effect estimation (see [1--3]).

Furthermore, while extreme weights may theoretically exist, I question whether the proposed method can fully address this issue. Consider a case where two populations are systematically different and share no overlap in their covariate or outcome distributions. In such a scenario, it is impossible to construct valid prediction intervals through density ratio weighting, as the true density ratio becomes infinite or extremely large. This raises the question of whether WCP remains meaningful under such circumstances. Moreover, clipping the weights inevitably introduces bias, since it imposes artificial bounds where the true weights are unbounded. I suspect some form of bounded weight or compact covariate support assumption must already be implicit in the validity of WCP, which makes it unclear why weight clipping is treated as a remedy rather than as a fundamental regularity condition.

That said, I encourage the authors to provide rebuttals to my comments, as there may be some misunderstandings on my part. If the authors believe they can provide a strong case about the following aspects of their work, I am open to change my score and assessment: (i) the high novelty on techniques they used for weight clipping; (ii) the wide applicability and potential impact of this work in ML and statistics community as well as social/medical/other applications; (iii) no bias is principally introduced by weight clipping. If you believe these can be well addressed, please also consider my below other comments in rebuttal.

**Other comments**

- In simulated experiment, it looks like the covariates shift only considered the mean shift by beta parameter, but the variance/covariance has no shift? It may not be necessary (if you can justify your choice) but could also be interesting to have some supplemental results on varying the covariance parameter as well.

- The split conformal method does not account for covariate shift, so it is expected to perform poorly and produce biased results in your setting. I suggest emphasizing more clearly that its poor performance is expected rather than surprising. Although you briefly mention this point, it could be stated more explicitly. The reason I make this suggestion is that it may be somewhat unfair to treat this method as a direct competitor since it relies on stronger assumptions that are violated in your setup. However, it is still useful to include it for illustration (to demonstrate that methods ignoring covariate shift indeed fail in such scenarios). You just need to clarify that its inclusion is for illustrative purposes, not as a competing benchmark. As an analogy, in line 457 you wrote, “We did not evaluate LR-QR, as W was not compatible with the linear structed assumed in Joshi et al. (2025).” So you shouldn't be self-contracted by what you want and do not want to compare. (A side note: “structed” appears to be a typo and should be “structure.”)

- A follow-up point for split-conformal: while it is biased, its prediction efficiency is really high from the SDs of emipircal coverage rates. Please discuss why and is there a bias-variance trade-off between your method and split-conformal. When covariates shift is moderate or small, would split-conformal actually be a better practical alternative?

- Equation (4),  should the $\delta$ actually be small? If it is not too small, it doesn't seem the coverage is guarantee at a high probability and it is not meaningful.

- The statement of Assumption 3 may need some modification, "The bias is not too large" sounds vague and subjective. How large is large?

- Are Assumptions 3 and 4 strong in practice, especially Assumption 4 which does not even allow misspecification? I understand such condition could even be necessary in this type of problems and it is okay, I don't expect you relax it for this work, but is there any sensitivity analysis type argument, quantification for the bias by the misspecification or plain language justifications?

**References**

[1] Crump, Richard K., et al. "Moving the goalposts: Addressing limited overlap in the estimation of average treatment effects by changing the estimand." (2006).

[2] Ma, Xinwei, and Jingshen Wang. "Robust inference using inverse probability weighting." Journal of the American Statistical Association 115.532 (2020): 1851-1860.

[3] Ju, Cheng, Joshua Schwab, and Mark J. van der Laan. "On adaptive propensity score truncation in causal inference." Statistical methods in medical research 28.6 (2019): 1741-1760.

**Questions:**

No additional question to add here. See weaknesses.

---

> ### Author Response · Authors · 2025-11-21
>
> (1/2)
>
> We thank the reviewer for their thorough and critical feedback. We believe there are some misunderstandings regarding the novelty and contribution which we hope to clarify below. We have also uploaded a revised manuscript incorporating your feedback.
>
> **1. Novelty and implementation details:**
> While weight clipping is a known heuristic in causal inference (propensity trimming), applying it to Conformal Prediction requires a subtly different algorithm and a fundamentally different analysis. In particular, we are not aware of any prior literature giving the $L_1$-generalization guarantee which can be obtained by our **Theorem 1**. This specific guarantee is critical to our coverage guarantees in **Theorem 2 and Theorem 3**, which require the $L_1$-error of the estimated density ratio to converge to $\Delta_B$. In this sense, our main novelty lies in (i) the analysis of CLISF and (ii) connecting the error guarantee of CLISF to an interpretable, useful guarantee for downstream use in conformal prediction. In particular, this allows us to bypass the dependence on higher moments of $w^*$, which is the primary theoretical bottleneck in current literature for obtaining similar guarantees with WCP. We believe this is a theoretical contribution, not just an implementation detail.
>
> **2. Wider applicability:** We believe our methods to be particularly applicable in settings with heavy-tailed covariate shifts, such as medicine and finance. These are settings where prior methods cannot be applied due to restrictive assumptions on the boundedness of $w^*$, which will not hold in a heavy-tailed setting. In our revisions we have included a discussion of this in a new **Section 4.3**.
>
> **3. Does clipping introduce bias?** As the reviewer has pointed out, clipping naturally introduces bias because $w^*$ may exceed the clipping parameter on some areas. However, in our original submission we have already emphasized that such bias is inevitable; a key aspect of our work is that we provide a finite-sample mechanism to *correct* the bias (**Lemma 2**), which would have otherwise led to undercoverage. We additionally provide and validate strategies to select $B$ (so as to balance the bias $\Delta_B$ and variance) in **Section 3.2 and Appendix E**.
>
> The reviewer also asks if asks if conformal prediction can be meaningful when overlap fails (infinite weights). We believe the answer to this question is *yes*, depending on the amount of overlap. For example, as long as the total variation is small, the clipping bias $\Delta_B$ will be small and thus **Theorem 2 and Theorem 3** will still provide a useful guarantee. This is similar to our **Example 1**. On the other hand, the reviewer raises a valid point in that if the distributions $P$ and $Q$ are completely disjoint, then coverage is basically impossible. This is true not just for density ratio based algorithms, but any algorithm in our setting: to guarantee exact coverage, a conformal prediction method will need some method of gaining information about the labels of the parts of $Q$ which are not covered by $P$. In this setting, our **Theorem 2 and Theorem 3** will still achieve a valid coverage guarantee of $\geq 1- \alpha$, but may significantly overcover; but this is unavoidable as previously discussed.
>
> **3. Regarding the synthetic data experiments:**
> Our synthetic data experiments (Section 5.2) involve a shift in the mean, but the Needle example (Example 1), the iWildCam experiment (Section 5.1), and the Communities and Crime experiment (Appendix F.1) represent complex, non-mean shifts. The Crime dataset specifically involves shifts based on racial subgroup frequencies, which alters the joint distribution complexly. For Section 5.2, we only considered a shift in the mean was so we could describe the magnitude of the shift in a single parameter.
>
> **4. Regarding split conformal as a baseline and the bias-variance tradeoff:**
> We have revised our text to explicitly emphasize in Section 5 (Experiments) that split conformal has been included in the evaluation to illustrate the necessity of accounting for the covariate shift.
>
> The reviewer has also brought up a good point about the efficiency of split conformal, particularly for small covariate shifts (e.g., small values of $\beta$ in the synthetic experiment). It is true that split conformal has small undercoverage when the shift is small, and small variance because there is no reweighting. However, it is often desirable to have *no undercoverage*, which is achieved by **Theorem 2 and Theorem 3** but not by split conformal. Additionally, a key advantage of our framework is that by increasing the sample size, we allow for larger values of $B$ and hence reduce the bias from $\text{TV}(P, Q)$ to zero. This is a key motivation of the structural risk minimization-base approach we empirically validate in **Section 3.2 and Appendix F.2**.

---

> ### Author Response · Authors · 2025-11-21
>
> (2/2)
>
> **5. Regarding Assumption 3 (Bias is not too large):** $\Delta_B \le 1/2$ is purely for analytical convenience (to bound a denominator). In our revisions we have emphasized that we can replace $1/2$ with any value less than $1$, and only impact the final constants of **Theorem 3**. In practice, since we control $B$, we can usually ensure this holds by increasing $B$; discuss this in **Appendix E**.
>
> **6. Assumption of No Misspecification (Assumption 4):**
> You correctly pointed out that assuming $w^* \in \mathcal{W}$ is unrealistic. We have generalized our theoretical results. **Theorem 2** and **Corollary 1** in the revised paper now explicitly account for model misspecification error, denoted as $\Delta_R$. The bounds now include an undercoverage term scaling with $\sqrt{\Delta_R}$. This undercoverage term vanishes under the original assumption that $w^* \in \mathcal{W}$. We have also included a discussion of the dependence of our guarantees on the misspecification error ni **Section 4.2**.
>
> **7. Other clarifications:**
> - In equation (4), the $\delta$ parameter is a confidence parameter and should be thought of as small. Our guarantees hold for any $\delta > 0$.

---

> > ### Comment · Reviewer_Nxh1 · 2025-11-21
> >
> > The authors addressed most of my concerns and misunderstanding. The solution of Response 6 beyond my expectation - thanks, and the key issue about the novelty is clarified and I think the work seems to be non-trivial compared to some data-driven trimming approaches in causal inference, such as Ma and Wang [2].
> >
> > Final comments:
> >
> > For Response 2, Li et al. (2019) AJE paper cited in Section 4.3 may not be the best reference. This paper only briefly says the extreme weights exist instead of giving any concrete examples if I read correctly. The only related text I found from this paper may be "Extreme propensities are particularly common in the setting of “big data,” where inclusion criteria can be defined broadly. The increasing prevalence of large data sources precipitates the need to clarify best practice for handling extreme propensity scores" in Section 1. And this paper is mainly a simulation study and no real data examples.
> >
> > I think you can still extend your discussion for wide applicability. For example, some participants never receive a certain treatment, e.g., ICU patients, older or patients with severe situations cannot be assigned to placebo, etc. Please try to elaborate real examples and find other literature to support that part. Please consider highlighting the extreme density ratio issues in real-world applications upfront in abstract or introduction to better motivate the paper, and I suggest you do not use the Needle example (Example 1) is the key motivation because it is not a real dataset, but you can say that is an illustration and helps formalize the problem.
> >
> > For Response 5, the added discussion paragraph before Assumption 3, I think you can potentially do a better job by incorporating slightly more details from Appendix E. Please also write to refer readers to Appendix E for more details about Assumption 3 in the main text, and connect the texts in Appendix E with Assumption 3 better.
> >
> > These additional comments are minor in my view, so they do not prevent me from accepting the paper now. Hopefully the authors could consider these suggestions in the next version.

---

> > > ### Author Response · Authors · 2025-11-22
> > >
> > > We sincerely thank the reviewer for their re-evaluation and for raising their score. We are very encouraged that our revisions regarding model misspecification and the novelty of our theoretical analysis helped resolve your concerns.
> > >
> > > We appreciate your constructive suggestions for the next version of our manuscript and will incorporate them as follows:
> > >
> > > * **References and Applicability:** We agree that concrete clinical examples are more compelling than the theoretical illustration alone. We will supplement the Li et al. (2019) citation with literature documenting specific scenarios (such as ICU patients) to better highlight the issue of extreme density ratios in the introduction and abstract.
> > > * **Assumption 3:** We will revise the text preceding Assumption 3 to explicitly refer readers to Appendix E.
> > >
> > > Thank you again for the time and effort spent reviewing our paper!

---

### Official Review · Reviewer_QaPK · 2025-10-28

**Soundness:** 3
**Presentation:** 3
**Contribution:** 3
**Rating:** 6
**Confidence:** 3

**Summary:**

The paper is well-written, easy to read and follow the ideas.

Problem context: Conformal prediction methods guarantee coverage under the assumption of exchangeability between training and test data. However, this assumption often fails under covariate shift. While Weighted Conformal Prediction (WCP) reweights calibration samples using estimated density ratios, it can perform poorly when these ratios are unbounded or inaccurately estimated, leading to high variance and under-coverage.

Proposal: The authors propose Clipped Least-Squares Importance Fitting (CLISF) and the corresponding Clipped Weighted Conformal Prediction (CWCP) framework. The main idea is to clip the learned density ratios at a threshold $B$ to reduce variance and stabilize conformal prediction under unbounded or heavy-tailed shifts.

Theoretical results: The paper provides the finite-sample coverage guarantees for conformal prediction using clipped importance weights. Specifically, the authors derive (i) an $L_2$-generalization bound for the clipped density ratio estimator, (ii) a concentration result for estimating the clipping bias $\Delta_B$, and (iii) dataset-conditional and expected coverage guarantees for CWCP with sample complexity.

Methodology: The approach consists of three stages: (1) learning a clipped density ratio via CLISF over a bounded class $\mathcal{W}_B$, (2) estimating the clipping bias $\Delta_B$ from data, and (3) running weighted conformal prediction with a slightly inflated target coverage level $1 - \alpha + \widehat{\Delta}_B$. Theoretical bounds rely on Rademacher complexity, weighted DKW inequalities, and standard uniform convergence tools.


Experiment: Empirical results on synthetic and real-data settings validate the theoretical results.

Overall, the paper makes a technically solid contribution by providing a principled theoretical foundation for weight clipping in conformal prediction. While the core idea is conceptually simple and based on existing heuristics, the analysis is rigorous and insightful.

**Strengths:**

1. The paper is well-written, though mathematically dense, but the authors have taken care to elaborate on the key ideas underlying all theoretical results, making the technical content accessible.

2. The problem paper tackles is of significant practical importance. It provides a rigorous theoretical treatment of the clipping heuristics commonly used for stabilizing importance weights in conformal prediction frameworks under distribution shift.

**Weaknesses:**

1.  While the theoretical analysis elegantly characterizes the bias-variance trade-off governed by the clipping parameter $B$, and the paper may discuss strategies, the empirical validation appears to rely on evaluating $B$ over a fixed grid.

2. The proposed Clipped Least-Squares Importance Fitting (CLISF) procedure is based on the $L_2$-optimal objective of the standard LSIF. The subsequent theoretical results, particularly the generalization bounds (e.g., Theorem 1 on $L_2$ error), inherently rely on the well-established properties of this objective and its connection to Rademacher complexity for the chosen function class $\mathcal{W}$.

**Questions:**

1. Do the paper's core finite-sample, dataset-conditional coverage guarantees rely specifically on the CLISF (i.e., the $L_2$-optimal) density ratio estimation procedure, or are the theoretical results (Theorem 2) general enough to accommodate other clipped density ratio estimators? Would it require a complete re-derivation of the generalization bounds based on the new objective function or perhaps with minor modifications?

2. The paper reports coverage results. Beyond validity, how does the clipping parameter $B$ influence the expected size of the resulting prediction sets? Is there a principled way to select $B$ that balances the need for low clipping bias ($\Delta_B$) with the desire for small prediction set size?

3.  Can the authors empirically validate a practical procedure for selecting $B$?

---

> ### Author Response · Authors · 2025-11-21
>
> We thank the reviewer for the high-quality review and for highlighting the rigorous nature of our analysis. We appreciate the detailed summary and the constructive questions. We have uploaded a revised manuscript incorporating your feedback.
>
> **1. Dependence of results on CLISF vs. General Estimators:**
> This is an important distinction. **Theorem 2 and Theorem 3** are general. They hold for *any* clipped density ratio estimator $\hat{w}$, provided one can establish a bound on its $L_1$ error with respect to the clipped truth $w^\star_B$. On the other hand, **Theorem 1** is specific to CLISF. It provides the necessary finite-sample generalization bound to prove that $\hat{w}$ indeed converges to $w^\star_B$ without depending on the higher moments of the unclipped $w^\star$. Thus, if a user prefers a different estimator (e.g., a clipped kernel density estimator), they can substitute it into our theoretical framework, provided they can prove an $L_1$ generalization guarantee akin to Theorem 1.
>
> **2. Influence of $B$ on Prediction Set Size:** In **Appendix F.1 (Figure 4)**, we analyze the Communities and Crime dataset and report both coverage and **prediction set width**, although we did not provide commentary on the widths in the initial submission. We have added explicit commentary on the observed set sizes in **Appendix F.1**. We observed a trade-off: for a fixed sample size, very small $B$ (high bias) leads to tighter sets but undercoverage, while very large $B$ recovers WCP behavior (good coverage on average, but high variance and potentially very large sets due to unstable weights).
>
> **3. Practical Procedure for Selecting $B$ (Empirical Validation):** A principled approach to selecting the value of $B$ is the **structural risk minimization**-inspired approach proposed in **Section 3.2**. In our revised submission, we have performed an empirical investigation of SRM for clipped density ratio estimation in **Appendix F.2**. We show that determining $B$ via SRM closely tracks the optimal test-loss performance. Specifically, the SRM-regularized objective correctly identifies the optimal value of clipping parameter for varying sample sizes for a well-chosen regularization strength.

---

> > ### Comment · Reviewer_QaPK · 2025-11-22
> >
> > Thank you for the clear and detailed responses. The distinction between the estimator-specific bound (Theorem 1) and the more general clipped coverage guarantees (Theorems 2 and 3) is now much clearer to me now. The added discussion on prediction set widths and the SRM-based procedure for choosing $B$ also addresses my concerns about efficiency and practical tuning. These revisions improve the clarity and completeness of the paper.
> > I am raising my score accordingly. I have no further concerns.

---

### Official Review · Reviewer_888h · 2025-11-02

**Soundness:** 3
**Presentation:** 3
**Contribution:** 3
**Rating:** 6
**Confidence:** 3

**Summary:**

The paper proposes clipped least-squares importance fitting to address the issue of unbounded likelihood ratio in conformal prediction under covariate shift. Experiments on classification and regression demonstrate the effectiveness of the method.

**Strengths:**

1. Extensive theoretical analysis is presented in section 3 to prove the resultant coverage guarantee by the proposed approach.

2. Both classification and regression tasks prove the outperformance of the work.

3. Discussion of choosing proper parameter B in Section 4.3 is insightful.

**Weaknesses:**

1. 30 trials in Figure 1 seems insufficient with quite unsmooth CDFs.

2. A figure presenting Example 1 can intuitively show the issue of unbounded ratio.

3. Notations are used without introduction, such as \hat{w} in Line 48.

**Questions:**

1. Localized conformal prediction [1] aims to provide conditional coverage guarantee given X=x, and it also relies on density estimation. Can your work extend to that topic? 2. Can you explain in Line 179-180 the equation about \triangle B when B =1.

[1]Guan, Leying. "Localized conformal prediction: A generalized inference framework for conformal prediction." Biometrika 110.1 (2023): 33-50.

---

> ### Author Response · Authors · 2025-11-21
>
> We thank the reviewer for their positive assessment of our work, specifically for recognizing the extensive theoretical analysis and the insightfulness of the parameter selection discussion. We have uploaded a revised manuscript incorporating your feedback.
>
> **1. Regarding the 30 trials and CDF smoothness:**
> We acknowledge that the CDFs appear step-like. We only managed 30 trials due to the high computational cost of running the full end-to-end pipeline on the iWildCam dataset (training models + density ratio estimation + conformal calibration) repeatedly. However, we believe the current results already demonstrate the variance reduction of CWCP compared to WCP and other methods. As mentioned in the original submission, the ideal CDF would be a steep step function about $0.8$: this indicates (i) low coverage bias on average and (ii) low variance in coverage. We believe that even with only $30$ trials, one can see that CWCP (particularly with $B = 10$ and $B = 20$) most clearly satisfies this criterion (compared to WCP, split conformal, and LR-QR). We will also rerun the experiments with a greater number of trials to make this result even more convincing.
>
> **2. Visualization of Example 1:** We have added **Figure 1** to the revised paper (Page 2), which visualizes the ``Needle'' example. It explicitly shows how the density ratio blows up on a small region $\mathcal{B}$, driving the second moment to infinity while keeping the total variation distance small.
>
> **3. Undefined notations ($\hat{w}$):** We have corrected the manuscript to explicitly define $\hat{w}$ upon first use (Line 48); it refers to an estimated density ratio.
>
> **4. Extension to Localized Conformal Prediction (LCP):**
> To our understanding, localized CP (Guan, 2023) typically weights nonconformity scores by a localized kernel $w(x) \propto K(x, x_{new})$. Our method focuses on the covariate shift weight $w(x) = dQ/dP$. Thus, our method and Localized CP are solutions to different problems. In principle, these can be combined: one could use CLISF to learn a stable density ratio $\hat{w}_{clip}(x)$ and multiply it by the localization kernel for a ``Robust Localized CP.''
>
> **5. Explanation of $\Delta_B$ when $B=1$ (Lines 179-180):**
> When $B=1$, the clipping bias becomes $\Delta_1 = \mathbb{E}_P[(w^*(X) - 1)^+] = \frac{1}{2}\int |p(x)-q(x)| \ dx = TV(P,Q)$.

---

### Author Response · Authors · 2025-12-02
**Summary of Author-Reviewer Discussion for Area Chair**

Given the exceptional circumstances affecting this year's review process, we would like to provide a summary of our reviewer discussion period, during which we received constructive feedback and made significant improvements to our manuscript. We believe that the post-rebuttal discussion indicates strong post-discussion support for our work.

**Key improvements.** Following reviewer feedback, we uploaded a revised manuscript addressing the major concerns:

* **Addressed model misspecification** (Reviewer wftD's primary concern): We generalized Theorem 2, Theorem 3, and Corollary 1 to explicitly account for misspecification error in the hypothesis class, with bounds that include undercoverage terms scaling with this error.

* **Added empirical validation of the structural risk minimization-based parameter selection method** (Reviewer QaPK's request): We included new experiments in Appendix F.2 demonstrating that structural risk minimization successfully identifies optimal clipping parameters.

* **Enhanced practical applicability discussion** (Reviewer Nxh1's concern): We added Section 4.3 discussing heavy-tailed covariate shifts in real-world applications and clarified when our method provides advantages over alternatives.

* **Added efficiency metrics** (Reviewer wftD's request): We expanded Appendix F.1 with explicit commentary on prediction set sizes and the bias-variance tradeoff.

**Summary of rebuttal outcomes.** The discussion period revealed that reviewers' concerns were largely addressable through clarification and revision. As directly stated in the replies of reviewers QaPK and Nxh1, two reviewers explicitly indicated stronger support for acceptance after our rebuttal.

* Due to our clarifications about our  theoretical guarantees and our added experiments, reviewer **QaPK** (initial score: 6) indicated increased support for our paper:

    > Thank you for the clear and detailed responses. The distinction between the estimator-specific bound (Theorem 1) and the more general clipped coverage guarantees (Theorems 2 and 3) is now much clearer to me now. The added discussion on prediction set widths and the SRM-based procedure for choosing B also addresses my concerns about efficiency and practical tuning. These revisions improve the clarity and completeness of the paper. I am raising my score accordingly. I have no further concerns.

* Due to our clarifications about the novelty of our approach, practical applicability, and addressing model misspecification, reviewer **Nxh1** (initial score: 2) indicated that we had addressed all major concerns preventing them from accepting the paper:

  > The authors addressed most of my concerns and misunderstanding. The solution of Response 6 beyond my expectation - thanks, and the key issue about the novelty is clarified and I think the work seems to be non-trivial compared to some data-driven trimming approaches in causal inference, such as Ma and Wang [2].

  In particular, they mentioned a few minor points for the camera-ready version and concluded:

  > These additional comments are minor in my view, so they do not prevent me from accepting the paper now.

* While the other two reviewers 888h (initial score: 6) and wftD (initial score: 4) had not responded by the time that the discussion was abruptly closed, we believe that we have addressed almost all of their major concerns. In particular, for reviewer wftD we addressed the main concern about model misspecification, efficiency metrics, and the practical applicability of our methods.

We hope that this information helps as the AC writes their meta-review. We believe the discussion (particularly the reversal from reject to accept by Reviewer Nxh1) reflects the meaningful improvements made to our work and indicate strong post-discussion support for our work.

---

### Meta-Review · Area_Chair_biZM · 2026-01-06

**Summary:**

1. Novelty and Fundamental Contribution: The strongest concern, emphasized by reviewer Nxh1 (who recommended rejection), was the perceived lack of originality in weight clipping, which parallels established methods like propensity score trimming in causal inference. They argued that clipping does not fundamentally solve unbounded density ratios (e.g., in non-overlapping supports) and may introduce irreducible bias, framing the work as a minor implementation detail rather than a core advance in conformal prediction. While the other reviewers (888h and QaPK) appreciated the finite-sample guarantees as a step forward, they did not strongly counter this, noting the approach's reliance on specific assumptions and estimators. The authors' rebuttal defended the conformal-specific guarantees as novel, but this did not sufficiently differentiate from prior works.

2. Theoretical Generality and Dependence on Specific Estimators: Reviewer QaPK questioned whether the guarantees extend beyond CLISF to other clipped estimators, potentially limiting the framework's applicability. The authors clarified that Theorems 2 and 3 are general (requiring only $L_1$ error bounds), but Theorem 1 remains CLISF-specific. This response helped, but highlighted the method's narrower scope than initially implied, reinforcing concerns about over-reliance on a particular estimator and underscoring the incremental nature of the contribution.

**Reviewer Concerns:**

Addressed:

1. Technical and presentation issues such as insufficient trials leading to unsmooth empirical CDFs, lack of visualizations for examples like the "Needle" distribution, and undefined notations (e.g., $\hat{w}$): Addressed through commitments to rerun experiments with more trials, addition of a new Figure 1 for illustration, and corrections to define notations in the revised manuscript.
2. Extension to related methods like localized conformal prediction (LCP): Addressed by explaining orthogonality and potential for combination into a "Robust Localized CP."
3. Dependence of coverage guarantees on the specific CLISF estimator: Addressed by clarifying that Theorems 2 and 3 are general (requiring only L1 error bounds), while Theorem 1 is CLISF-specific, enhancing framework flexibility.
4. Reliance on grid search for B and need for principled selection: Addressed via an empirical study in Appendix F.2 using structural risk minimization (SRM), showing it effectively tracks optimal performance across sample sizes.

Outstanding:

1. Lack of high novelty in weight clipping techniques: methods like propensity score trimming in causal inference were not sufficiently differentiated, leaving the work seen as incremental rather than transformative.

2. Inability to fully resolve unbounded density ratios in extreme cases (e.g., non-overlapping distributions): introducing irreducible bias not directly countered, despite mentions of data-estimable bias correction via $\Delta_B$.
Broader implications of bias-variance trade-offs in non-ideal or worst-case scenarios: added appendices provide empirical insights but lack new theoretical bounds for efficiency and robustness.

**Reviewer Scores:**

Reviewer 888h and QaPK may maintain their scores, while Reviewer Nxh1 may increase to 4.

---

### Decision · Program_Chairs · 2026-01-26

Reject